



# Joint Cloud Water Path and Rain Water Path Retrievals from ORACLES Observations

Andrew M. Dzambo[1,2], Tristan L'Ecuyer[1], Kenneth Sinclair[3,4], Bastiaan van Diedenhoven[5], Siddhant Gupta[2], Greg McFarquhar[2], Joseph R. O'Brien[6], Brian Cairns[3], Andrzej P. Wasilewski[3,7], Mikhail Alexandrov[3,5]

[1] University of Wisconsin – Madison, Madison, WI, USA
[2] University of Oklahoma – Cooperative Institute for Meteorological Mesoscale Studies (CIMMS), Norman, OK, USA
[3] NASA Goddard Institute for Space Studies, New York
[4] Universities Space Research Association (USRA), Columbia, MD 21046, USA
[5] Columbia University, New York
[6] University of North Dakota, Grand Forks, ND 58202, USA
[7] SciSpace LLC

*Correspondence to*: Andrew M. Dzambo (dzamboam@ou.edu)

## Abstract

This study presents a new algorithm that combines W-band reflectivity measurements from the Airborne Precipitation Radar - 3[rd] generation (APR-3), passive radiometric cloud optical depth and effective radius retrievals from the Research Scanning Polarimeter (RSP) to estimate total liquid water path in warm clouds and identify the contributions from cloud water path (CWP) and rainwater path (RWP). The resulting CWP estimates are primarily determined by the optical depth input, although reflectivity measurements contribute ~10-50% of the uncertainty due to attenuation through the profile. Uncertainties in CWP estimates across all conditions are 25% to 35%, while RWP uncertainty estimates frequently exceed 100%.

Two thirds of all radar-detected clouds observed during the ObseRvations of Aerosols above CLouds and their intEractionS (ORACLES) campaign that took place from 2016-2018 over the southeast Atlantic Ocean have CWP between 41 and 168 g m$^{-2}$ and almost all CWPs (99%) between 6 to 445 g m$^{-2}$. RWP, by contrast, typically makes up a much smaller fraction of total liquid water path (LWP) with more than 70% of raining clouds having less than 10 g m$^{-2}$ of rainwater. In heavier warm rain (i.e. rain rate exceeding 40 mm h$^{-1}$ or 1000 mm d$^{-1}$), however, RWP is observed to exceed 2500 g m$^{-2}$. CWP (RWP) is





found to be approximately 30 g m$^{-2}$ (7 g m$^{-2}$) larger in unstable environments compared to stable environments. Surface precipitation is also more than twice as likely in unstable environments. Comparisons against in-situ cloud microphysical probe data spanning the range of thermodynamic stability and meteorological conditions encountered across the southeast Atlantic basin demonstrate that the

combined APR-3 and RSP dataset enable a robust joint cloud-precipitation retrieval algorithm to support future ORACLES precipitation susceptibility and cloud-aerosol-precipitation interaction studies.

## 1 Introduction

Stratocumulus clouds (StCu here after) are the dominant cloud type over Earth's subtropical oceans.

Significant advances in our understanding of subtropical StCu have been documented in many previous studies including their typical cloud fractions [Stephens et al., 2012; King et al., 2013; L'Ecuyer et al., 2019], radiative impacts [Hang et al., 2019] and associated precipitation processes using data from past field experiments [Stevens et al., 2003], long-term ground-based data collection [Mann et al., 2014; Yang et al., 2018], and spaceborne remote sensors [Fox and Illingworth, 1997; L'Ecuyer and Stephens, 2002;

Lebsock et al., 2011; Chen et al., 2011; Lebsock et al., 2011; Douglas and L'Ecuyer, 2019; Werner and Deneke, 2020]. Despite these advances, until recently observations of cloud and precipitation processes over the southeast Atlantic Ocean were very limited.  The StCu cloud deck over the southeast Atlantic Ocean is influenced by a biomass-burning aerosol layer from July through October [Zuidema et al., 2016], complicating our understanding of global radiation and climate impacts [Redemann et al., 2020 and

references therein]. Prior to the beginning of the ObseRvations of Aerosols above CLouds and their intEractionS (ORACLES) experiment, the 2000 SAFARI field campaign was the only other airborne field campaign to collect measurements of the expansive cloud deck over the southeast Atlantic [Haywood et al., 2003]. Such measurements of the southeast Atlantic StCu deck are critically needed, given the general lack of process-level understanding of cloud-aerosol-precipitation processes [Hou et al., 2018; Mülmenstädt et

al., 2020] and the consequent uncertainties they propagate in modern-day global climate models [e.g. Stephens et al., 2010; Sockol and Small Griswold, 2017; Cesana et al., 2019].

Two main themes often emerge from cloud-aerosol-precipitation studies: (1) disentangling cloud-aerosol-precipitation processes from meteorological controls is perhaps the biggest challenge [Zhou et al., 2015; Douglas and L'Ecuyer, 2019; Douglas and L'Ecuyer, 2020]; and (2) partitioned cloud and

precipitation datasets are especially limited and must be developed using robust retrieval techniques with well-quantified uncertainties [Lebsock et al., 2011; Lebsock and L'Ecuyer, 2011; Mace et al., 2016; Cadeddu et al., 2020]. Many studies have shown that both aerosols and precipitation change cloud morphology and vertical radiation or heating profiles [e.g. Albrecht 1989; Feingold et al., 1996; Wood



2005; Nelson et al., 2016; Nelson and L'Ecuyer, 2018; Zhang and Zuidema, 2019; Abel et al., 2020; Painemal et al., 2020], and hence accurate quantification *and* collocation of macrophysical cloud properties is required. The solution for a collocated synergy of measurement platforms was met during ORACLES. Although cloud-aerosol-precipitation interactions are not the primary focus of this study, this work is

highly motivated by the need for synergy between ORACLES measurement platforms in order to address ORACLES Level 2 and 3 science objectives [Redemann et al., 2020] involving aerosol semi-direct effect [Koch and Del Genio, 2010; Das et al., 2020] and aerosol indirect effects [e.g. Twomey 1977; Albrecht 1989; McFarquhar and Wang, 2006; Diamond et al., 2018].

       Understanding the interplay between cloud and precipitation processes is critically important to the

design and development of experiments and frameworks for comparisons between modeling and observation platforms [Mace et al., 2016; Witte et al., 2019]. Simultaneous retrievals of cloud and precipitation properties from spaceborne remote sensors have enhanced our understanding of global cloud-aerosol-precipitation interactions [e.g. L'Ecuyer and Stephens, 2002; Lebsock and L'Ecuyer, 2011; Mace et al., 2016], and similar algorithms adapted for airborne platforms can reinforce our understanding through

eliminating some inevitable limitations present using spaceborne sensors such as, for example, improved spatiotemporal data resolution [Dzambo et al., 2019], helping to fulfill a fundamental need to construct datasets required to validate spaceborne remote sensing algorithms and further explore ORACLES science objectives.

       We develop and test a joint radar-radiometer CWP and RWP retrieval using W-band reflectivity

profiles from the Airborne Precipitation Radar – 3$^{rd}$ Generation (APR-3) and cloud properties from the Research Scanning Polarimeter (RSP), both of which flew aboard the NASA P-3 aircraft during all of ORACLES. This algorithm (referred to as WCOD hereafter) is analogous to that described in Lebsock and L'Ecuyer (2011) and updated from the algorithm described in Dzambo et al. (2019). Both CWP and RWP are evaluated using available in-situ cloud probe measurements. The next section describes each dataset in

more detail, with the subsequent sections describing the algorithm mechanics, typical and limiting cases for this algorithm, and evaluation of these retrieved variables.

## 2 Datasets

       The APR-3 and RSP flew aboard the NASA P-3 during ORACLES, allowing high-resolution

profiles of rainwater content and cloud water path to be simultaneously retrieved. Cloud and precipitation properties retrieved from these instruments are evaluated using available in-situ cloud probe datasets described in Section 2.3.



## 2.1 The Advanced Precipitation Radar – 3$^{rd}$ Generation (APR-3)

The APR-3 measured profiles of collocated Ku- (13 GHz), Ka- (35 GHz) and W- (95 GHz) band reflectivity and Doppler velocity, revealing cloud vertical structure and precipitation properties in unprecedented detail [Dzambo et al., 2019; hereafter D19]. For ORACLES, the W-band reflectivity

channel is used exclusively because the Ka-band and Ku-band's sensitivities were too low to detect many of the observed StCu decks. The APR-3 W-band channel has a sensitivity between -35 and -38 dBZ at a range of 1 km. Under typical operating configurations at high altitude (approximately 7 km), which represent the bulk of the data used for this study, the W-band channel has sensitivity of around -30 dBZ. The effects of non-uniform beamfilling and multiple scattering are minimal, given that the radar has a

narrow beamwidth of 0.9 degrees. Over the course of the three ORACLES campaigns, the APR-3 collected over 18 million reflectivity profiles with vertical resolutions ranging from 35 meters to 8.6 meters depending on the radar's operational configuration. During ORACLES 2016, over 3 million reflectivity profiles at each frequency were collected primarily in very stable, non- or lightly-precipitating regions off the coast of Namibia while close to 7 million profiles were collected in 2017 in marginally stable

environments containing more inhomogeneous clouds with both convective and stratiform warm rain south of Saõ Tomé. In 2018, nearly 8 million reflectivity profiles were collected sampling mostly homogeneous cloud with some convective rain near the equator, noting very similar sampling environments as in 2017. A comparison of WCOD retrieval data between 2016 and 2017 is discussed in much greater detail in D19, and a brief overview of precipitation frequency during each campaign year is presented in Redemann et al.

(2020). Finally, surface noise or "ground clutter" is limited to about 200 meters (sometimes less) in the W-band observations. More details about the APR-3 observations during ORACLES are described in Dzambo et al. (2019).

## 2.2 The Research Scanning Polarimeter (RSP)

The Research Scanning Polarimeter (RSP) is an airborne version of the Earth Observing Scanning Polarimeter (EOSP) [Travis 1992; Cairns et al., 2003]. The RSP makes polarimetric and total intensity measurements in 9 spectral bands ranging from the visible to shortwave infrared. The RSP has a 0.8 deg field-of-view and uses along track scanning to make up to 152 measurements every 0.84 seconds. The measurements sweep approximately ±60° from nadir along the aircraft's track. Cloud top height is

derived from the RSP observations using a multi-angle parallax method [Sinclair et al., 2017]. Subsequently, RSP data are mapped so that multi-angle views are available as a function of location at cloud top [Alexandrov et al., 2012].



The RSP retrieves properties of the cloud droplet size distribution at cloud top, namely the effective radius ($r_e$) and effective variance ($v_e$, not used in this study), using the polarized reflectances of the cloudbow in the scattering angle region between 135 and 165 degrees [Alexandrov et al., 2012]. The single scattering properties of cloud (or rain) droplets are directly linked to the shape of the cloudbow, which

enables a simple retrieval of cloud properties using Mie calculations [Alexandrov et al., 2012]. Cloud optical depth (COD) is retrieved using visible reflectances [cf. Nakajima and King, 1990].

For this work, multi-angle polarimetry is advantageous because polarimetric retrievals rely on the shape of the cloudbow, not its intensity, and are therefore unaffected by above-cloud aerosol layers and cloud 3-D effects [Alexandrov et al., 2012]. The multi-angle measurements can robustly detect optically

thin clouds and the presence of multiple cloud layers [Sinclair et al., 2017]. For the ORACLES experiment, RSP measurements offer reliable data to constrain retrieved cloud water path. COD and $r_e$ retrieved from the 0.865 μm measurements are used, with $r_e$ corresponding to the cloud top and assumed through the depth of the profile (implications of this limitation are described in the next section). Although we assume $r_e$ through the depth of the profile, we note that the polarimetric $r_e$ corresponds to about 1-2 optical depth

within cloud top, which typically corresponds to a depth of 50-100 m [Alexandrov et al., 2012]. Finally, the RSP $r_e$ has recently been validated to within 1 μm against a Cloud Droplet Probe [Alexandrov et al., 2018] with typical values between 5 and 15 microns during ORACLES [see Figs. 5 and 9 in Miller et al., 2020].

The uncertainty associated with each instrument is more thoroughly described in Section 3 but a few important notes of caution regarding interpretation of the optical depth and $r_e$ data are presented here:

1.  Optical depth retrievals for very bright clouds have low accuracy because reflectance is almost saturated at its semi-infinite value [Nakajima and King, 1990]. Very bright reflectances exceeding the limits of the look-up table employed in the algorithm further reduce the accuracy of estimated LWP. The frequency of this occurrence during ORACLES is low and is estimated at 0.5%.

2.  During the 2016 deployment, aerodynamic interferences with other instruments on the NASA P-3

25          aircraft caused the scanner of the RSP not to function properly, leading to extensive data loss (see Table 1). For this paper, only data that are not substantially affected by this issue were used.

3.  Cloud retrievals from RSP are only available when the P-3 aircraft was flying above cloud top and the solar-viewing geometry was such that the scattering angles required for the polarimetric drop size retrievals were sampled.

## 2.3 Cloud Probes

### 2.3.1 Overview



The CWP and RWP retrievals from the APR-3 and RSP data are evaluated against the cloud CWP and RWP derived using the droplet number distribution (i.e. n(D)) sampled by a suite of in-situ cloud probes. In-situ sampling of the marine stratocumulus was conducted during vertical cloud profiles [Gupta et al., *in preparation*] when the aircraft ascended or descended through the cloud layer at a vertical velocity of

around 5 m s$^{-1}$. Data from the in-situ probes were averaged to 1-second resolution, and hence the droplet n(D) was available at a vertical resolution of 5 meters. The in-situ cloud probes available during ORACLES included components of the Droplet Measurement Technologies (DMT) Cloud Aerosol and Precipitation Spectrometer (CAPS; comprised of a Cloud and Aerosol Spectrometer or CAS, a Cloud Imaging Probe or CIP, and a DMT hot-wire probe), the DMT Cloud Droplet Probe Version 2 (CDP) and

the Stratton Park Engineering Company (SPEC) Two-Dimensional Stereo Probe (2DS). Each of these instruments are briefly described in the next subsections, with more details on each instrument given in Gupta et al. *in prep*., Redemann et al. [2020], and references therein.

### 2.3.2 Instruments

The CAS and the CDP sample particle n(D) as a function of particle diameter D by measuring the intensity of light emitted by a laser which is then scattered over some scattering angles (between 4-12°) by particles passing through the probe's sample volume [Baumgardner et al., 2001; Lance et al. 2010]. The CAS and CDP data were both processed using the University of North Dakota Airborne Data Processing and Analysis Software Package (ADPAA), an open source software [Delene et al. 2011]. Both the CAS and

CDP can measure particles with 0.5 (2) < D < 50 μm and particles with D > 3 μm are identified as cloud droplets in this study. CAS data were available for most of the research flights across the three deployments (Table 1) and therefore, CAS data are used for n(D) over 3 < D < 50 μm. For research flights starting from 10 October 2018, the CAPS components had an instrument malfunction and CDP data were used. The CDP has been deployed in numerous field campaigns and has been used for evaluation of cloud properties from

various remote sensing platforms [e.g. King et al., 2013; Alexandrov et al., 2018; Sarkar et al., 2020].

The SPEC 2DS is used to sample droplet n(D) beyond the detection limit of the CAS and CDP, i.e., for D > 50 um. The 2DS is an optical array probe (OAP) that consists of two arrays with 128 photodiodes (mounted both horizontally and vertically) where each photodiode has an effective pixel size of 10 microns [Lawson et al. 2006]. The droplet n(D) is determined as each droplet passing through the probe sampling

volume occludes a certain number of photodiodes proportional to its diameter and produces a two-dimensional shadow image. The 2DS is a stereo probe with a horizontally (H) and a vertically (V) mounted arrays measuring droplet n(D) simultaneously and independently. However, due to soot deposition on the inside of the receive side optical lens for the V channel, only data from the H channel data were available



for ORACLES 2016. To maintain consistency, 2DS n(D) from the H channel is used for 2017 and 2018 as well, despite the availability of n(D) from the V channel for these deployments. The 2DS data were processed using the University of Illinois/Oklahoma Optical Probe Processing Software [UIOOPS; McFarquhar et al. 2018] and were used to determine droplet n(D) for 50 < D < 1280 um.

The droplet n(D) from the CAS and CDP (D < 50 um) as well as 2DS (50 < D < 1280 um) were combined and droplet concentration ($N_c$), $r_e$ and LWC were calculated for each 1-second average. LWC was calculated by integrating the droplet mass distribution over the entire size range ($3^{rd}$ moment of the droplet size distribution) and a 1-second average is recorded as a cloud sample if $N_c > 10$ cm$^{-3}$ and LWC > 0.05 g m$^{-3}$. The LWP was calculated for each cloud profile by integrating the LWC for each cloud sample

from cloud base to cloud top. Since droplets with D > 50 um are typically classified as drizzle, LWP calculated using LWC over the 3 < D < 50 um (D > 50 um) size range was classified as the in-situ CWP (RWP). A comparison of in-situ CWP and RWP and retrieval-based CWP and RWP is presented in Sect. 5. Uncertainties of probe-derived CWP and RWP are unavailable at this time, but acknowledge the sizing uncertainty in CAS/CDP sizing is approximately 10% with corresponding N uncertainties of less than 50%.

## 3 Retrieval Methodology

     The joint APR-3 and RSP cloud and precipitation retrieval algorithm (hereafter, denoted WCOD) is an extension of the radar-only rainfall algorithm described in Dzambo et al. (2019). This updated algorithm explicitly accounts for cloud water, constrained using the RSP COD estimate (sensitive to cloud water and

rainwater contents), which provides additional insights into how cloud- and rain- water contents are distributed through each radar profile. Such partitioning has been done before with ground-based radar [e.g. Cadeddu et al., 2020; Wu et al., 2020] and satellite observations [Lebsock and L'Ecuyer, 2011; Mace et al., 2016]. An airborne analogue to this partitioning method is presented in this section.

### 3.1 Retrieval Mechanics

     The mechanics of the algorithm presented here closely follow those outlined in Lebsock and L'Ecuyer (2011), where CloudSat W-band radar reflectivity measurements and a cloud optical depth constraint from MODIS are used to infer precipitation rate and cloud water path. The only difference between the WCOD approach and Lebsock and L'Ecuyer (2011), aside from the data sources, is the

explicit use of a cloud-top effective radius. The WCOD algorithm uses a radar reflectivity profile ($\mathbf{Z}$), a COD ($\tau$) and path-integrated attenuation (PIA) constraints:

$$\mathbf{Y} = [Z_1, Z_2, \ldots, Z_N, \tau, PIA] \quad (1)$$





to solve for a profile of rainwater contents or rainfall rates (**R**) and a cloud water path (CWP):

$$X = [R_1, R_2, \dots, R_N, CWP] \quad (2)$$

where the subscript N represents the total number of reflectivity measurements, and $R_N$ is either the rainwater content or rainfall rate at the surface. The algorithm attempts to minimize a cost function:

$$\Phi(X, X_a, Z) = \left[[F(R) - Z]^T S_Z^{-1}[F(R) - Z]\right] + \left[(X - X_a)^T S_a^{-1}(X - X_a)\right] + \frac{(\tau_{mod} - \tau)^2}{\sigma_\tau^2}$$

$$+ \frac{(PIA_{mod} - PIA)^2}{\sigma_{PIA}^2} \quad (3)$$

where **X** represents the retrieved precipitation rates and CWP, $X_a$ represent the a priori precipitation rates

and CWP (described in section 3.3), while $\tau_{mod}$ and $PIA_{mod}$ represent the modeled optical depth and path integrated attenuation respectively. The last two terms on the right hand side of Eq. 3 are the integral constraints from the observed COD and PIA. The visible optical depth observed by the RSP includes contributions from CWC and RWC, i.e.:

$$\tau_{mod} = \frac{3Q_{ext}}{4\rho_w} \int_{H_{bot}}^{H_{top}} \left(\frac{CWC}{r_{e,cld}} + \frac{RWC}{r_{e,pcp}}\right) dz \quad (4)$$

On the right hand size of Eq. 4, $\rho_w$ is the density of liquid water, $Q_{ext}$ is the extinction efficiency (set

to 2 in the limit of geometric optics) and $r_{e,cld}$ and $r_{e,pcp}$ are the effective radii corresponding to the cloud and rain water contents. The cloud effective radius in this case is directly input from available RSP data, whereas the rain effective radius is parameterized following Abel and Boutle (2012) via a look-up table. Assumptions about the vertical variation of CWC and $r_{e,cld}$ are discussed in section 3.2. Equation 4 supplies the physical model for the optical depth constraint in Eq. 3. The algorithm mechanics described in

this subsection are further illustrated in Fig. 1.

### 3.2 Observation and model uncertainties

The available data from ORACLES enables finer tuning of several key assumptions made in D19. First, PIA is estimated by taking the difference between σ0 in a given radar profile and the observed surface

backscatter (or σ0) in a nearby clear-sky profile. This reduces the need for the look-up table (LUT) of clear-sky σ0 and has the benefit of fixing the measurement uncertainty to that of the APR-3 measurement uncertainty (1 dB). For most radar data, clear sky PIA estimates are possible using this technique; however, the LUT approach is still needed for estimating PIA and PIA uncertainty for scenes with extensive unbroken cloud cover. Second, the vertical resolution of the radar and the added τ constraint allow CWP to

be appropriately distributed through the observed cloud. Retrieving a complete profile of CWC requires vertically resolved cloud measurements insensitive to precipitation, which are not generally available from





remote sensing instrumentation. Nevertheless, reasonable CWC estimates are necessary to constrain attenuation due to cloud water. In Lebsock and L'Ecuyer (2011) and D19, cloud water content was parameterized as a function of cloud top height (H) and surface rain rate (R) (i.e. Eq. 10 in Lebsock and L'Ecuyer, 2011):

$$log_{10}(CWP) = 2.147 + 0.011H + 0.132log_{10}(R) \ [Nimbostratus]$$
$$log_{10}(CWP) = 2.186 + 0.017H + 0.129log_{10}(R) \ [Congestus]$$

and CWC was assumed to be constant with altitude. The algorithm introduced here instead assumes that the cloud base altitude occurs at the altitude of maximum reflectivity, a characteristic of precipitating warm clouds, and CWP is distributed from the top of the cloud (i.e. the first radar reflectivity) down to this altitude following Bennartz (2007) who suggested that cloud water content (CWC) increases proportionally to height (H) vertically, i.e.:

$$CWC \propto H \quad (5)$$

$$CWP = \int_{H_{bot}}^{H_{top}} CWC(z)dz \quad (6)$$

where $H$ is the in-cloud altitude.

Preliminary results from ORACLES show that CWC increases with height approximately following Eq. 5, though entrainment near the cloud top can result in CWC becoming constant or decreasing rapidly with altitude at the top of the cloud [Gupta et al., *in prep*]. During ORACLES, the maximum radar reflectivity sometimes occurs at or near cloud top, meaning the entire CWP would be distributed into the

top 1-3 radar bins and result in unreasonably large CWCs. To circumvent this potential problem, CWP is distributed through the top 6 radar bins if the cloud spans six or more bins (6 bins corresponds to approximately 200 meters of cloud depth from radar profiles taken at transit altitudes), or spans the entire profile if the cloud spans fewer than 6 bins. Otherwise, CWP is distributed from the top of the cloud to the altitude of maximum reflectivity if the maximum radar reflectivity is greater than -15 dBZ, which several

studies have suggested is representative of drizzle onset [Mace and Sassen, 2000; Liu et al., 2008], and assumed to be cloud base in this study. This methodology is very similar to the methodology presented by Wu et al. (2020), ensuring all CWC occurred above cloud base. We also note that the retrieved CWP is insensitive to the number of radar bins used for inferring CWC in the column (not shown).

We account for uncertainty in the CWC profile inferring CWP using the following two

approximations:

$$CWP = \gamma_{ad}\tau_c\rho_w r_e \quad (7), and$$
$$CWP = \gamma_{const}\tau_c\rho_w r_e \quad (8)$$





where $\tau_c$ is the optical depth from the cloud water component, $\rho_w$ is the density of liquid water, $r_e$ is the cloud top effective radius, and $\gamma_{ad}$ in Eq. 7 is valid for cases when CWC increases with altitude in cloud (i.e. the adiabatic assumption, with $\gamma_{ad} = 5/9$), whereas $\gamma_{const}$ is valid if the cloud has a vertically invariant drop distribution (i.e. $\gamma_{const} = 2/3$). These models introduce uncertainty of about 20% in the

CWP estimates. The RSP effective radii contribute additional uncertainties of 5% to the CWP retrieval [Alexandrov et al., 2018]. Reliable uncertainty estimates of COD in regimes with high reflectance are not available at this time, and are not accounted for in this version of the algorithm.

With this model for the vertical distribution of cloud water, the error covariance matrices in Eq. (3) are constructed as

$$S_y(Z) = \sigma_{Z\_meas}^2 + \sigma_{att}^2 + \sigma_{Z\_DSD}^2 \quad (9),$$
$$S_y(\tau) = \sigma_{\tau\_meas}^2 + \sigma_{cwp}^2 + \sigma_{r_e}^2 \quad (10), and$$
$$S_y(PIA) = \sigma_{PIA\_meas}^2 \quad (11)$$

where $\sigma_{Z\_meas}^2$, $\sigma_{att}^2$ and $\sigma_{Z\_DSD}^2$ are the uncertainties from the measured reflectivity, modeled attenuation and drop size distribution assumptions respectively, $\sigma_{\tau\_meas}^2$ is the uncertainty in the measured COD, $\sigma_{cwp}^2$ is the uncertainty in the CWP estimate between Eqs. 7 and 8, $\sigma_{r_e}^2$ is the uncertainty in $r_e$, and $\sigma_{PIA\_meas}^2$ is the uncertainty in the measured PIA. In the present work, we adopt the Abel and Boutle (2012) raindrop size distribution and retain the 2 dB uncertainty in reflectivity adopted by Lebsock and L'Ecuyer (2011) to

allow for potential overestimation of the error from the DSD. Though it is beyond the scope of the present study, collocated Ka-band and Ku-band radar channels, could be used in the future as a constraint on DSD parameters (e.g. slope and shape parameters).

### 3.3 A Priori Estimate

The a priori constraint serves to keep the algorithm from retrieving unrealistic precipitation rates, RWP and CWP, while also providing a reasonable initial estimate of each. As in Lebsock and L'Ecuyer (2011), the full profile of rainfall rates is set to an a priori value of 0.1 mm h[-1] with a variance that extends 3 orders of magnitude around this value (i.e. allowing retrieved rainfall rates to vary between $10^{-4}$ and $10^{2}$ mm h[-1]).

The a priori constraint for CWP is estimated using an adjusted adiabatic water model:

$$CWP_a = \int_{0.8H}^{H} z f_{ad} \, \Gamma_{ad} dz \quad . \quad (12)$$

The terms $z$, $f_{ad}$ and $\Gamma_{ad}(T, p)$ represent the altitude in-cloud (where H is equal to the distance between cloud top and cloud base following our definitions), the degree of adiabacity (1 = fully adiabatic, anything





less is sub-adiabatic) and the adiabatic increase of liquid water content with height. Following Merk et al. (2016) the terms $f_{ad}$ and $\Gamma_{ad}$ are set to 0.8 and 2.0 g m$^{-3}$ km$^{-1}$ respectively, although in practice $\Gamma_{ad}$ is a function of both pressure and temperature and can vary between 0.3 and 1 [Albrecht et al., 1990; Merk et al., 2016]. Computing the exact value using the local thermodynamic state offers no additional value to the

5    algorithm given the large uncertainty associated with the a priori estimate. Using the full radar profile for the a priori CWP estimate following this method would result in extremely large and unrealistic CWP, thus, the a priori CWP is computed using the radar bins closest to the cloud top. Finally, in the absence of RSP data, the a priori CWP and distribution of CWC through the radar profile defaults to Eq. 10 in Lebsock and L'Ecuyer (2011); this procedure replicates CWC as in the first version of WCOD (i.e. 2C-RAIN described

10    in D19).

### 3.4 Contribution Matrices

To assess the relative contribution of each source of uncertainty to the retrieved CWP and rainfall rates at each level, the contribution matrix (or C-matrix) is computed. Following Lebsock and L'Ecuyer

15    (2011)

$$\boldsymbol{S}_x = \boldsymbol{C}_a + \boldsymbol{C}_Z + \boldsymbol{C}_\tau + \boldsymbol{C}_{PIA} \text{ , (13)}$$

$$\boldsymbol{C}_a = \boldsymbol{S}_a^{-1}, \quad (14)$$

$$\boldsymbol{C}_Z = K^T \boldsymbol{S}_Z^{-1} K \text{ , (15)}$$

$$\boldsymbol{C}_\tau = \frac{L^T L}{\sigma_\tau^2} \text{ , and (16)}$$

$$\boldsymbol{C}_{PIA} = \frac{M^T M}{\sigma_{PIA}^2} \quad (17)$$

where K, L and M are partial derivatives of reflectivity, optical depth and PIA with respect to prescribed perturbations in reflectivity, COD and PIA respectively. Each C-matrix provides the fractional contribution of each error source to the uncertainty in all retrieved quantities. The improved uncertainty characterization by virtue of having high-resolution data from ORACLES offers a unique opportunity to assess the impact

20    of each C-matrix on the final retrieved quantities and their respective uncertainties. A similar method was employed in Leinonen et al. (2018), where a multi-frequency radar technique was developed for snowfall and demonstrated the utility of ensuring their retrieved quantities were well constrained by the observations. The contributions of each algorithm input are discussed extensively in Section 4.



## 4 Algorithm Performance and Results

A total of 1.28 million collocated RSP and APR-3 profiles containing valid RSP COD and $r_e$ and APR-3 detected cloud cover span all 3 ORACLES deployments. Retrieved surface precipitation rates greater than 1 mm h⁻¹ (or 24 mm d⁻¹) make up only 3% of the total dataset. However, trade cumulus near

the Equator and cumulus in the StCu-to-Cu transition region around Ascension Island often contained surface precipitation rates greater than 1 mm h⁻¹. To demonstrate the algorithm's performance across common conditions observed during ORACLES, while also highlighting the algorithm's performance in a limiting case such as trade cumulus precipitation, two representative case studies are selected and presented in the following two subsections. A summary of CWP and RWP statistics for the entire ORACLES

campaign is also presented.

### 4.1 Drizzling Stratocumulus

Figure 2 shows a typical drizzling stratocumulus scene from ORACLES. This scene, which spans a little over one minute of flight time (or ~5 km), contains two drizzling cells with maximum column rain

rates reaching a maximum value approaching 1 mm d⁻¹. Nearly every profile contains a RWP between 0.1 and 10 g m⁻². The scene shown in Fig. 2 also shows a nearly uniform CWP and total optical depth. As noted previously, the RSP performs best over non-broken cloud, and given that most total LWPs are under 150 g m⁻² in this scene, the retrieved CWPs are very accurate with uncertainties between 25-30%.

The C-matrix computations and mean precipitation rate uncertainties (Fig. 3) for the weakly

drizzling scene in Fig. 1 are given in Fig. 4. The uncertainty contributions to the retrieved precipitation rate profile come almost entirely from the reflectivity. This finding follows both Lebsock and L'Ecuyer (2011) and Lebsock et al. (2011), which found that CloudSat rainfall retrievals nearly exclusively rely on the observed reflectivity profile in the retrieved precipitation rate profile for weakly (or non-) drizzling stratocumulus. The mean uncertainties near cloud top in this scene reflect weak precipitation on the order

of ~10⁻² mm h⁻¹ or less. The evaporation model results in complete evaporation of precipitation before reaching the surface, which is expected given the mean rain drop radii near the surface were consistently less than ~40 μm (not shown). The C-matrix computations for CWP reveal contributions between 0.4 and 0.55 by the observed reflectivity with the remaining contribution coming almost entirely from the observed optical depth.

### 4.2 Trade Cumulus





The scene in Fig. 2 represents a very typical case from ORACLES, for which the WCOD algorithm was optimized. To illustrate the limitations of this algorithm, we examine a heavily precipitating trade cumulus observed by the APR-3 along a NW-to-SE flight track just south of the Equator (Fig. 5). The observed W-band reflectivity exceeded 20 dBZ in many of the profiles, and experienced heavy attenuation

exceeding 20 dB during the first minute of the scan. During this period, the algorithm corrects the near-surface reflectivity (from the lowest resolvable bin up to ~700 meters) by 15-25 dB. For several profiles, the propagation of errors by the algorithm results in non-retrievable solutions indicated by missing data in the center of Fig. 5.

Observed optical depths by the RSP often exceeded 20. The heaviest precipitation within the first

minute of the scan, evident in the modeled reflectivity plot (Fig. 5, top-middle panel) and corroborated by the RWP and RR retrieved quantities, results in rain optical depths approaching 5 (bottom panel). Corresponding RWP retrievals fall in the ~300-500 g m$^{-2}$ range with the heaviest rain containing nearly 800 g m$^{-2}$ of RWP. The CWPs associated with these heaviest rain cells are below ~100 g m$^{-2}$ suggesting that a substantial fraction of cloud droplets have been converted to rain. Data contained in the 2$^{nd}$ minute of the

scene in Fig. 5 tell a much different story. Total optical depths in this time range generally vary between 5 and 10 units, but the much lower RWC and less than 1 dB of PIA contained in these profiles results in the total LWP being dominated by CWP. Noting the lower reflectivity in the 2$^{nd}$ half of this segment, the much larger CWP relative to RWP is indicative of less active collision-coalescence. Broken cloud is evident between minutes 1.4 and 1.6, which introduce a potential source of uncertainty in the RSP COD via 3-D

effects.

The measurement and a priori contributions to the retrieved rainfall rate at each level and retrieved CWP are shown in Fig. 6. From the main panel of Fig. 6, the reflectivity profile contributes the most to the final retrieved precipitation profile, with the a priori constraint contributing ~10-20% of the uncertainty in the bottom half of the profile. PIA contributes a small but non-negligible amount of uncertainty in the

lowest bins, consistent with results shown in Lebsock and L'Ecuyer (2011). The PIA contribution often exceeds 5-10% for the largest rainfall rates (not shown) and, unsurprisingly, adds further uncertainty to near-surface rainfall rates [L'Ecuyer and Stephens, 2002]. Near the top of the cloud, where reflectivity exclusively contributes to the uncertainty, the retrieved precipitation rate uncertainties are ~100%. Uncertainties increase with depth in the cloud reaching 300-1000% in the bottom of each profile consistent

with growing uncertainty due to the accumulated errors in attenuation corrections. At the base of the cloud, uncertainties with any appreciable PIA contribution (i.e. > 0.01) result in uncertainties exceeding 1000% or more. Clearly even successful retrievals in the first minute of this scene have rainfall rates near the upper limit of the range of algorithm applicability consistent with prior spaceborne radar rainfall retrievals [e.g. Haynes et al, 2009].





Fig. 6 also shows that the a priori constraint contributes heavily in heavier rainfall. Eliminating such profiles would leave only profiles where the final uncertainty was determined by the PIA, reflectivity and optical depth. Figure 7 shows how the profile of mean precipitation rate uncertainty changes when the PIA and a priori constraints are eliminated from the pool of available profiles. The mean contribution from the a

priori constraint is < 0.05 at all levels in this scene. However, removing profiles with the a priori constraint (any profile with an a priori contribution > 0.01) results in a reduction of uncertainty by factors of 2-5. Eliminating the contributions of PIA to the total uncertainty yields a minor reduction to the mean uncertainty profile but is not nearly as dramatic as when the a priori contribution is removed. When comparing the uncertainties between Fig. 6 and Fig. 7, the uncertainty in the bottom half of the heavily

precipitating profiles scale down by factors of 2-3 compared to the weakly drizzling profiles. This difference is almost assuredly due to the large attenuation propagating uncertainties deeper into the cloud [Hitschfield and Borden, 1954]. Comparing Fig. 6 and Fig. 3 also reveals less contribution by the PIA especially near the surface. This could be due to one of the following two reasons:

   1.  The PIA integral constraint is not linked to the retrieved cloud water path as in Lebsock et al.

15       (2011), and/or

   2.  The PIA uncertainty is manifested in the reflectivity uncertainty through the $\sigma_{att}^2$ term, thereby distributing the (little) attenuation through the column.

     Optical depth contributes approximately 90% to the total uncertainty in CWP, with the reflectivity contributing the remaining 10%. This contribution by reflectivity is significantly less than that observed by

the drizzling stratocumulus scene, presumably because attenuation effects through the profile exacerbate uncertainty at each altitude through the cloud. Similar to the drizzling stratocumulus scene, the a priori uncertainty does little to affect the final uncertainty in CWP, with most uncertainties falling between 25-30%. The a priori constraint contributes very little to the retrieved CWP indicating that cloud water is well constrained by the available observations.

25       These two test studies reveal the following general characteristics of the WCOD algorithm:

   1.  For cases when optical depth is appreciable and moderate to heavy precipitation is falling, the CWP retrieval relies more on the observed optical depth because large RWP affects the uncertainty in the optical depth more than the reflectivity, and reflectivity is subject to both attenuation and DSD uncertainties that propagate deeper into a profile.

2.  For cases in weakly or non-drizzling StCu, the uncertainty in the optical depth becomes larger relative to the uncertainty in the reflectivity due to less DSD and attenuation uncertainty (even though RWP is small), resulting in a larger contribution by the reflectivity.





3. Regardless of the case, the measured optical depth consistently contributes a majority (i.e. between 0.5 and 0.9 in these two cases) of the total uncertainty in any CWP retrieval.

**5 WCOD CWP and RWP Evaluation**

**5.1 Evaluation Methodology**

As with any airborne-based evaluation study between in-situ and remotely sensed observations, a big uncertainty is the fact that the in-situ and remote sensing instrumentation never actually measure the same volume of cloud at the same time. This is especially difficult for precipitation, where comparing precipitation estimates is challenging due to collocation issues [e.g. Meneghini et al., 2001], small-scale

heterogeneity along short distances [e.g. Tridon et al., 2019] and general complications arising from diagnostics used for comparing precipitation variables between datasets [Kay et al., 2018]. Another issue arises with temporal comparisons of cloud properties. While the StCu cloud deck can exist for several days or longer, individual clouds grow and dissipate on time scales of less than an hour. To accommodate the resulting sampling mismatches, we collectively examine statistics for all quasi-simultaneous retrieval and

in-situ measurements across each individual campaign year excluding flights or data subsets where any of APR-3, RSP or cloud probes data are missing. We define quasi-simultaneous as any radar profile occurring within 5 minutes *and* within 0.25 degrees latitude or longitude (~25 km) of an in-situ profile. For a typical aircraft speed of 80 m/s, and noting the APR-3 collects 24 profiles every 1.2 seconds, this corresponds to a maximum of 12,000 comparable profiles per in-situ profile. For our study, this definition of quasi-

simultaneous allows for ~1,200 WCOD profiles per in-situ profile.

To ensure profiles from cloud probe data represent scenes that would be detected by the APR-3, any probe data measuring a cloud thinner than 200 meters (consistent with the majority of cloud depths estimated by radar in D19) or with a RWP less than 0.25 g m$^{-2}$ are not considered in this analysis. The number of profiles following this screening procedure, along with the total number of WCOD profiles

nearby a cloud probe profile following the aforementioned screening procedure, are shown in Table 2.

**5.2 Comparison between in-situ and retrieved CWP and RWP**

A summary of CWP and RWP retrieved and measured in-situ is presented in Fig. 8. RWP overall tends to agree somewhat better than CWP between WCOD and the in-situ estimates. CWP data for 2018

agree best, with the mean (median) CWP exceeding the probe estimates by 9 g m$^{-2}$ (8 g m$^{-2}$). The mean CWP for the WCOD and cloud probe data from each data subset are 79 g m$^{-2}$ and 70 g m$^{-2}$ respectively. The large standard deviations in CWP from WCOD suggest wide variability in cloud macrophysical





properties, especially given the 3-fold sampling increase. With this in mind, the variability in CWP is lowest in 2016 compared to either 2017 or 2018 for both WCOD and probe datasets, suggesting more similar cloud properties during this campaign year. We speculate CWP would agree better if all thin, broken cloud were observed, such that we could re-perform this verification technique without RWP or

cloud thickness thresholds.

The mean RWP from the WCOD and cloud probe data subsets is 15 and 9 g m$^{-2}$ respectively. RWP tends to be a fraction of the total LWP overall in most cases with the largest RWPs again coming in 2018 for both data subsets. Variability in RWP between WCOD and the cloud probes is lowest for 2016, corresponding to the low variability in CWP for 2016 as well.

Although RWP generally agrees between measurement platforms, several sources of uncertainty will need to be accounted for in future studies. Horizontal variability is one such source of CWP uncertainty, especially in-situ CWP estimates where each profile was computed over at least ~5 km geometric distance. Although we can rule out the possibility of ultra thin cloud biasing our analysis using this verification technique, the WCOD reflectivity screening procedure possibly removes some valid cloud

cover near the cutoff of the stated thickness threshold. Figure 9 supports this hypothesis, as several cloud probe estimates exceeding 15 g m$^{-2}$ (a proxy for raining conditions) occur at relatively low CWP.

We also recognize the bulk validation presented here does not necessarily imply point measurements between the cloud probes and nearest retrievable RWP from radar measurements are the same. The point of this evaluation was to compare the large-scale retrieval capabilities from remote sensing

platforms to accurate in-situ based measurements, and our effort to match WCOD profiles within a short period of time (5 min) and within a reasonable distance (25 km) help mitigate some uncertainty with this comparison. In any case, the collocation of CWP and RWP with above-cloud aerosol data from instruments like the HSRL promises a large, enriching dataset for the study of aerosol semi-direct and indirect effects. Our estimates of CWP and RWP relative to total LWP are reasonable and expected with all aspects

considered, and is supported by the recent Cadeddu et al. (2020) study (see Table 2, Table 4, and Fig. 9) which showed CWP dominates the total LWP signal in northeast Atlantic StCu.

## 6 Campaign Statistics of CWP and RWP

Throughout the ORACLES campaign, the APR-3 radar operators frequently reported situations where the aircraft was flying over StCu cloud yet the APR-3 radar failed to detect cloud cover. We find that, for everywhere a valid RSP COD and APR-3 profile were collocated, an estimated 45% of cloudy scenes appeared as "clear sky" to the APR-3. We showed in the previous section that, while controlling for





cloud thickness and RWP representative of drizzling StCu, CWP from APR-3 detected clouds are biased high compared to in-situ estimates. Further, evidence suggests that several ultra thin, broken and drizzling clouds are screened out of this analysis. The topic of APR-3 warm cloud detectability will be the topic of a future paper. The campaign statistics presented here, therefore, represent primarily cloudy profiles that are

either drizzling/raining or near the onset of precipitation.

The median CWP and RWP retrieved from the resulting scenes are 90.3 and 1.7 g m$^{-2}$ respectively [Table 3]. For CWP, the 1-σ range (68% of values) is between 41 and 168 g $^{-2}$ while the 3-σ range (99% of values) is between 6 and 445 g m$^{-2}$. RWP, by contrast has a 1-σ range (3-σ range) of 0.6 to 7 g m$^{-2}$ (0.1 to 2892 g m$^{-2}$). The range of WCOD retrieved values demonstrate that the majority of LWP across all clouds

is predominately CWP, although some heavy precipitation cases can dominate the total LWP. Recall, however, that this CWP range corresponds only to predominately precipitating clouds detected by the APR-3, with both ranges likely being lower if ultra thin cloud were detected and/or not screened out.

As noted in many previous studies, any meaningful analysis of cloud-aerosol-precipitation interactions must account for environmental variability [Douglas and L'Ecuyer, 2019; Douglas and

L'Ecuyer, 2020]. Two common measures of environmental state include estimated inversion strength (EIS; Wood and Bretherton, 2006) and sea surface temperature (SST). EIS is computed as:

$$EIS = LTS - \Gamma_m^{850}(z_{700} - LCL) \quad (18)$$

where LTS is the lower tropospheric stability (defined as the difference in potential temperature between 700 mb and the surface), $\Gamma_m^{850}$ is the moist adiabatic lapse rate at 850 hPa and LC is the lifting condensation level. We use the European Centre for Medium-Range Weather Forecasts (ECMWF) Re-Analysis (ERA)

Interim dataset [Dee et al., 2011] for SST and for computing EIS. Nearly all EIS values from ORACLES fell between -2 K and 12 K, while SSTs ranged from 289 K near the Namibian/Angolan coast to 301 K near the Equator. CWP, RWP, and maximum column precipitation rate all increase with increasing SST or decreasing EIS.

Table 3 reveals that both CWP and RWP were larger in unstable compared to stable environments.

There were nearly 30 times as many profiles collected in stable environments (i.e. EIS > 0 K) compared to unstable environments, demonstrating that (with a few exceptions) the majority of collocated remote sensing data collected during ORACLES sampled the StCu deck. The statistics in Table 3 are further broken down in Fig. 10. The probability of maximum reflectivity is between -20 and -15 dBZ for stable environments, whereas bimodal peaks in reflectivity occur in unstable environments with one peak between

-15 and -10 dBZ indicative of light drizzle or non-precipitating cloud and a second peak between 10 and 20 dBZ indicative of warm rain. This result is consistent with the cumulative frequency by altitude diagram in Fig. 8 of Dzambo et al. (2019).





Both Table 3 and Fig. 10 show that CWP in unstable environments tends to exceed CWP in stable environments for all values of maximum reflectivity. RWP, by contrast, is comparable in all environments for maximum reflectivities under 0 dBZ but again tends to be larger in unstable environments at maximum reflectivities above this value. The largest divergence in regime-based median RWPs occurs at reflectivities

> 10 dBZ, which occurs near the maximum range of valid W-band reflectivity measurements. This reflects the fact that CWP and RWP are sensitive to cloud thickness, and thicker clouds prevailed more often in unstable and marginally-stable (EIS between 0 K and 4 K) environments (Fig. 7 in D19). CWP values in excess of 150 g m$^{-2}$ are measured frequently in all environments but are especially frequent in unstable environments.

A cloud containing a maximum reflectivity of -15 dBZ or greater (i.e. the onset of drizzle) will have a RWP of at least ~1.5 g m$^{-2}$. Conditional precipitation fractions computed from the WCOD dataset reveal that about 28% of all cloudy profiles observed during ORACLES (detected by either the radar or the RSP) contained precipitation with surface precipitation occurring around 9% of the time. The precipitation fraction depends strongly on stability with precipitation occurring nearly twice as often in unstable

environments compared to stable environments (Fig. 11). Surface precipitation occurs less than 20% of the time in unstable environments, and less than 10% of the time in stable environments.

We emphasize that these statistics, with noted exceptions, include primarily drizzling scenes with some raining scenes. In addition to not detecting and/or screening out thin clouds, the APR-3 was often turned off in cases when extended clear sky periods occurred. Lidar measurements offer another

independent measure of cloud fraction, especially in situations where RSP data are unavailable. Thus, while this study focuses primarily on thicker clouds detected by the radar, these results should be combined with those from synergistic remote sensing instruments to completely characterize cloud and precipitation properties during ORACLES. Finally, although measurements and retrievals from the APR-3 or RSP do not suffer from aerosol-induced biases, cloud modifications via aerosol interactions remains an active topic

of research. Most of the biomass-burning aerosol observed during ORACLES was located in the free troposphere [Diamond et al, 2018; Pistone et al., 2019; Das et al., 2020], meaning aerosol semi-direct effects almost certainly occurred. Changes to the thermodynamic structure of the atmosphere – beyond the EIS analysis presented here – will be required to disentangle other potential environmental controls on measured and retrieved cloud properties.

**7 Conclusions**



A joint cloud-precipitation retrieval algorithm for retrieving collocated cloud and rainwater paths is developed to address ORACLES science objectives related to aerosol indirect and semi-direct effects. CWP retrieved by WCOD is relatively insensitive to the a priori assumption and completely insensitive to hydrometeor attenuation (PIA). For lightly drizzling scenes, the profile of reflectivity constrains CWP

nearly as equally as the input COD, whereas in heavily precipitating scenes CWP is much more strongly influenced by the COD integral constraint. Uncertainties in RWP are often a factor of 2-5 less in drizzling scenes than in heavily raining scenes. This work demonstrates the importance of collocated radar-radiometric measurements for partitioning total LWP into CWP and RWP, but especially reveals how much the reflectivity constraint affects retrieved CWP.

CWP and especially RWP are found to be higher in unstable environments (EIS < 0 K). Since most ORACLES data were collected in stable environments (EIS > 0 K), RWP is generally low throughout all three campaigns, although RWP values of 1.5 g m$^{-2}$ – a proxy for the onset of drizzle based on our analysis – occur in 54% of all collocated RSP/APR-3 profiles (28% when accounting for clear APR-3 profiles, consistent with Fig. 10). When the maximum column reflectivity exceeds 10 dBZ, RWP values quickly

grow to become a significant fraction of the total optical depth. RWP values of 15 g m$^{-2}$ are a good indicator of rain and occur 9% of the time in WCOD cloudy profiles.

Despite limited available in-situ data for comparison in 2016 and 2017, all retrieved median RWP estimates are within 4 g m$^{-2}$ of RWP estimates from probe data. CWP in both WCOD and probe estimates are largest for 2018 (70 and 62 g m$^{-2}$ respectively) and relatively constant between 2016 and 2017 (70 vs.

63 g m$^{-2}$ for WCOD, 32 vs. 27 g m$^{-2}$ for the probe estimates). RWP, by contrast, is lowest in 2016 in both probe (0.6 g m$^{-2}$) and WCOD (0.7 g m$^{-2}$) estimates. The 2016 ORACLES campaign year took place during the peak of the climatological biomass-burning season, so the possibility exists that the lower RWPs could be the result of aerosol indirect effects. The WCOD CWP and RWP standard deviations are largest for 2018, likely owing to some transitioning StCu-to-Cu and potentially (generally) cleaner clouds observed in

2017 and 2018. These data motivate future aerosol-cloud interaction and StCu-to-Cu transition studies, especially since our analysis illustrates how collocated CWP and RWP estimates and collocated in situ and lidar-based aerosol measurements may reveal insights into aerosol semi-direct and aerosol indirect effects. The encouraging agreement with in situ probes further motivates wider application of WCOD to a number of other recent airborne field campaigns with similar instrumentation (e.g. the recent CAMP$^2$EX

campaign).

**Data Availability**





The WCOD dataset can be accessed through the NASA ESPO archive via
https://espoarchive.nasa.gov/archive/browse/oracles/P3/WCOD. All data, including the APR-3, RSP and
cloud probe datasets, can be found at https://espoarchive.nasa.gov.

## 5   Competing Interests

The authors declare that they have no conflicts of interest.

## Author Contributions

AMD and TSL originated the main ideas and experiment designs. AMD created all tables and figures in the
text. KS, BVD, BC, AW and MA contributed the RSP data and advised/edited the RSP-related areas of the
text. SG, GM and JO contributed the cloud probe data. AMD prepared this manuscript with contributions
from all co-authors.

## Acknowledgements

The lead author would like to acknowledge his dissertation committee from the University of Wisconsin –
Madison for their support and feedback on the original versions of the work presented here: Grant Petty,
Dan Vimont, Michael Morgan and Sam Stechman. This work was funded under NASA Grant
NNX15AF99G.

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



**Tables**

**Table 1**: Description of data availability for the ORACLES 2016-18 field campaigns. The 27 Sep. 2016 flight was classified a "transit flight" but is included in this analysis because the transit flight took place entirely in the experiment area between Walvis Bay, Namibia and Ascension Island and is denoted by a single asterisk (*). No ER-2 data are included in this study. Y denotes available data for a given flight, whereas N indicates no data available. For the cloud probes (CPs), the instruments used to derive CWP and RWP are listed.

|  | APR-3 2016 | RSP 2016 | CPs 2016 |  | APR-3 2017 | RSP 2017 | CPs 2016 |  | APR-3 2018 | RSP 2018 | CPs 2018 |
|---|---|---|---|---|---|---|---|---|---|---|---|
| 08-30-16 |  |  |  | 08-12-17 |  |  |  | 09-27-18 | Y | Y | CAS, 2DS |
| 08-31-16 | Y | Y | N | 08-13-17 | Y | Y | CAS, 2DS | 09-30-18 | Y | Y | CAS, 2DS |
| 09-02-16 | Y | N | N | 08-15-17 | Y | Y | CAS, 2DS | 10-02-18 | Y | Y | CAS, 2DS |
| 09-04-16 | Y | N | N | 08-17-17 | Y | Y | CAS, 2DS | 10-03-18 | Y | Y | CAS, 2DS |
| 09-06-16 | Y | N | CAS, 2DS | 08-18-17 | Y | Y | CAS, 2DS | 10-05-18 | Y | Y | CAS, 2DS |
| 09-08-16 | Y | N | CAS, 2DS | 08-19-17 | N | N | CAS, 2DS | 10-07-18 | Y | Y | CAS, 2DS |
| 09-10-16 | Y | Y | CAS, 2DS | 08-21-17 | Y | Y | CAS, 2DS | 10-10-18 | Y | Y | CDP-2, 2DS |
| 09-12-16 | Y | N | CAS, 2DS | 08-24-17 | Y | Y | CAS, 2DS | 10-12-18 | Y | Y | CDP-2, 2DS |
| 09-14-16 | Y | Y | CAS, 2DS | 08-26-17 | Y | Y | CAS, 2DS | 10-15-18 | Y | Y | CDP-1, 2DS |
| 09-18-16 | Y | Y | CAS, 2DS | 08-28-17 | Y | Y | CAS, 2DS | 10-17-18 | Y | Y | CDP-2, 2DS |
| 09-20-16 | Y | Y | CAS, 2DS | 08-30-17 | Y | Y | CAS, 2DS | 10-19-18 | Y | Y | CDP-2, 2DS |
| 09-24-16 | Y | N | CAS, 2DS | 08-31-17 | Y | Y | CAS, 2DS | 10-21-18 | Y | Y | CDP-2, 2DS |
| 09-25-16 | Y | N |  |  |  |  |  | 10-23-18 | Y | Y | CDP-2, 2DS |
| 09-27-16* | Y | N |  |  |  |  |  |  |  |  |  |





**Table 2**

**Table 2:** Summary of median, mean and standard deviation values of CWP and RWP statistics from the
WCOD retrieval and cloud probe datasets. The mean and 1-sigma CWP or RWP are given in parentheses.
5     The top 3 rows include the number of WCOD-only profiles that were collocated to the in-situ cloud probe
data. The collocation procedure is described in the text.

| | WCOD | CLOUD PROBES |
|---|---|---|
| **# PROFILES (2016)** | 24,497 | 9 |
| **# PROFILES (2017)** | 60,418 | 15 |
| **# PROFILES (2018)** | 74,149 | 76 |
| **CWP (2016)** | 69.8 (78.4 ± 38.2) [g m$^{-2}$] | 31.7 (40.2 ± 23.2) [g m$^{-2}$] |
| **CWP (2017)** | 62.9 (78.9 ± 55.6) [g m$^{-2}$] | 27.2 (35.3 ± 19.2) [g m$^{-2}$] |
| **CWP (2018)** | 69.6 (89.3 ± 58.8) [g m$^{-2}$] | 61.6 (80.6 ± 63.3) [g m$^{-2}$] |
| **RWP (2016)** | 0.7 (3.1 ± 51.1) [g m$^{-2}$] | 0.6 (0.8 ± 0.5) [g m$^{-2}$] |
| **RWP (2017)** | 1.1 (5.9 ± 59.7) [g m$^{-2}$] | 3.2 (3.7 ± 3.5) [g m$^{-2}$] |
| **RWP (2018)** | 1.4 (30.1 ± 212.1) [g m$^{-2}$] | 4.6 (11.3 ± 18.9) [g m$^{-2}$] |





**Table 3**

**Table 3:** Summary of CWP and RSP statistics from the joint APR-3 and RSP joint retrieval algorithm. Values of CWP, RWP and maximum reflectivity are reported as median values (mean values in parentheses).

|  | # PROFILES | CWP [g m$^{-2}$] | RWP [g m$^{-2}$] | MAX REFL. [dBZ] |
|---|---|---|---|---|
| STABLE (EIS > 0 K) | 1,235,982 | 88.9 (105.3) | 1.6 (45.1) | -14.0 (-12.0) |
| UNSTABLE (EIS < 0 K) | 43,520 | 129.7 (144.2) | 8.1 (121.8) | -4.5 (-1.7) |
| ALL | 1,279,568 | 90.2 (106.7) | 1.7 (47.8) | -13.8 (-11.6) |





**Figures**

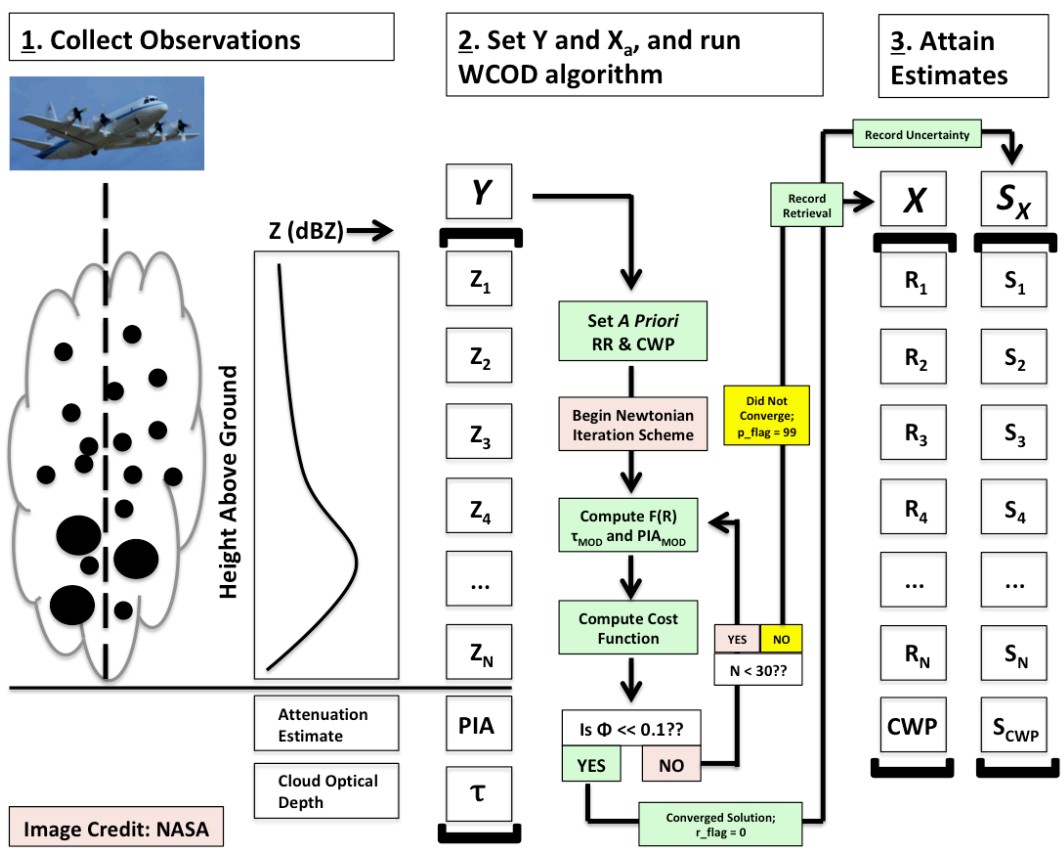

**Figure 1:** A diagram highlighting the algorithm mechanics described in Section 3 of the text. For the final retrieval vector **X** and corresponding uncertainties $S_x$, both rainfall rate and rainwater content are retrieved at each level. CWP is retrieved directly from this procedure, while RWP is computed by integrating the column RWC and adjusted following the sub-cloud evaporation procedure described in the text.



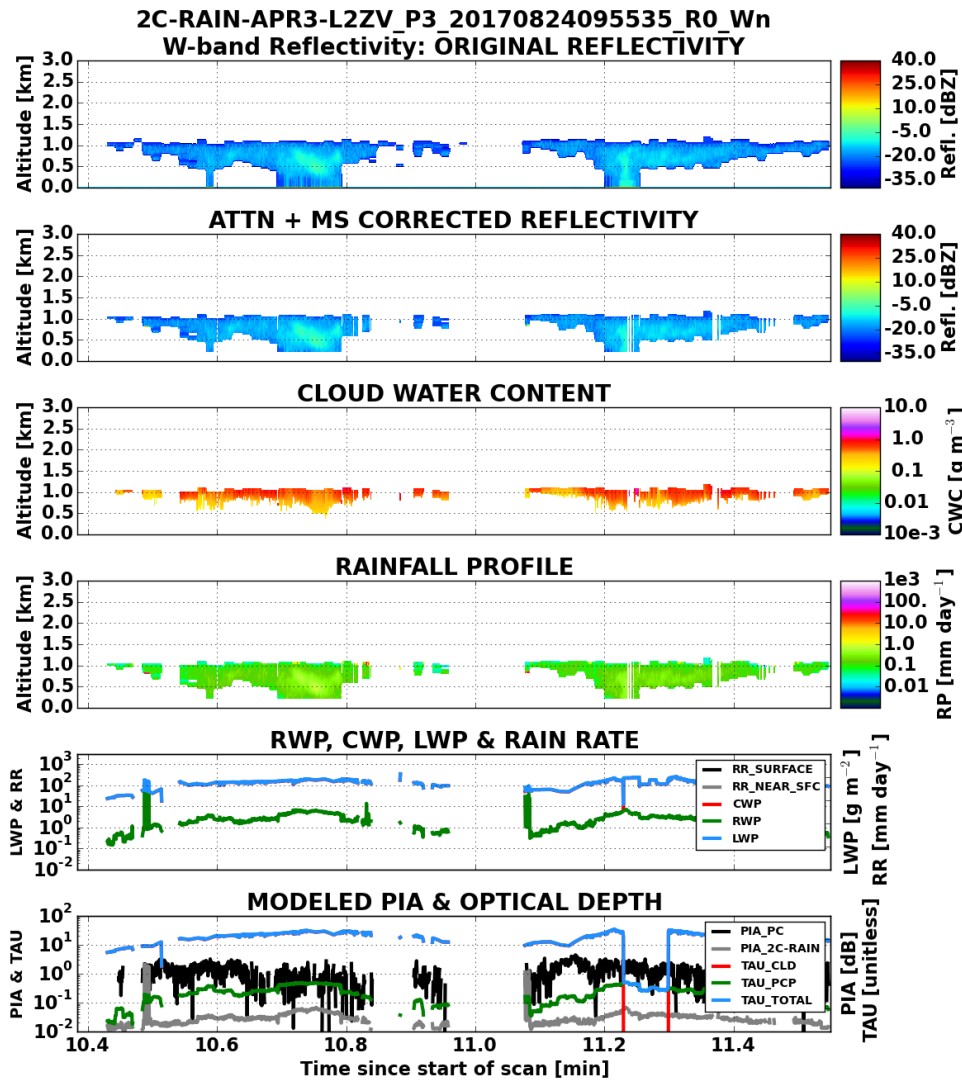

**Figure 2:** WCOD retrieval for a pair of lightly drizzling stratocumulus clouds observed by the APR-3 (top) on 24 Aug. 2017. The model-corrected reflectivity (top-middle) for this scene, along with retrieved RWC (top-center) and rainfall profile (bottom-center), are also shown. The bottom two panels show retrieved water paths (cloud, rain and total), rainfall rates (surface or evaporation-corrected rainfall rate and near-surface rainfall rate), attenuation ('PC' or observed and 'WCOD' or modeled) and optical depths (cloud, rain and total). The lower LWP and optical depth values around minute 10.5 correspond to a short period where the RSP was not operating.



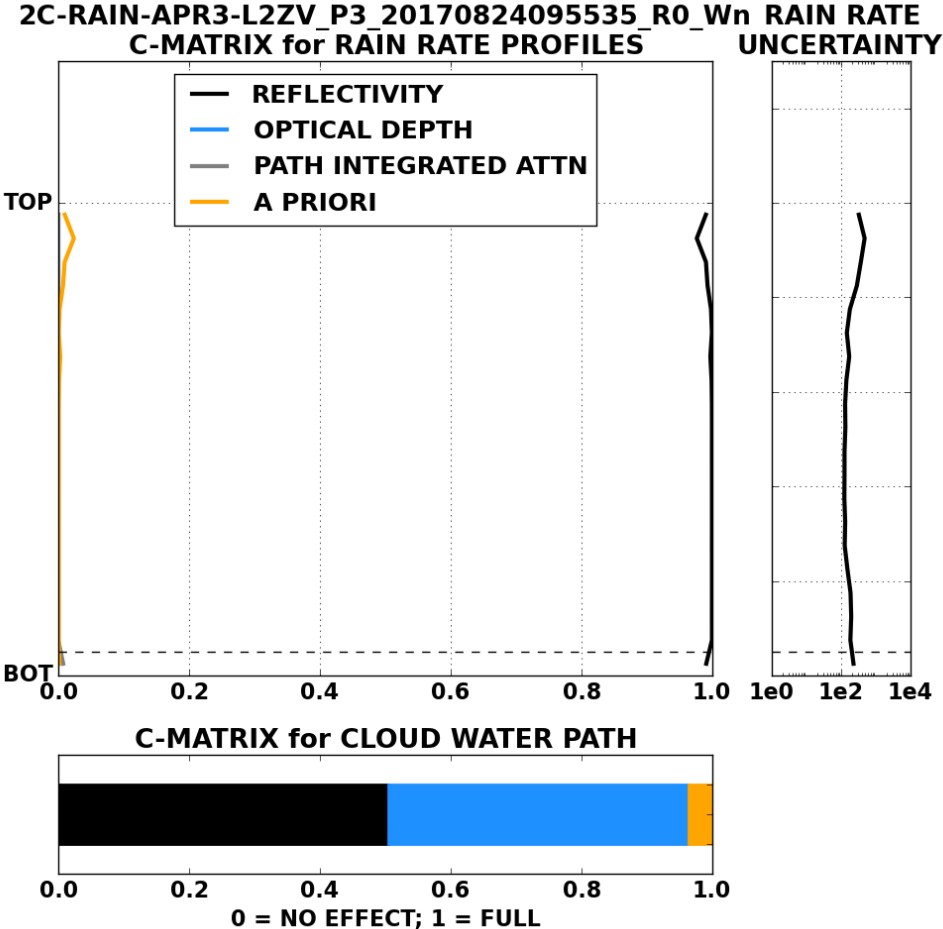

**Figure 3:** Mean contribution matrix (or C-matrix) calculations relative to cloud height for the radar profiles shown in Fig. 2. The C-matrix for each retrieved rainfall rate, as well as CWP, was calculated according to Eq. 13. Individual C-matrix profiles are normalized to the full length of the reflectivity profile (i.e. 0 = radar echo base, 1 = radar echo top). Data below the dashed line indicates a C-matrix calculation for both the surface (i.e. corrected for evaporation) and near-surface precipitation rates and is exactly the same for both precipitation rate quantities. In the bottom panel, the contributions from reflectivity, optical depth, PIA and a priori uncertainties are respectively shown in black, blue, gray and orange.

...





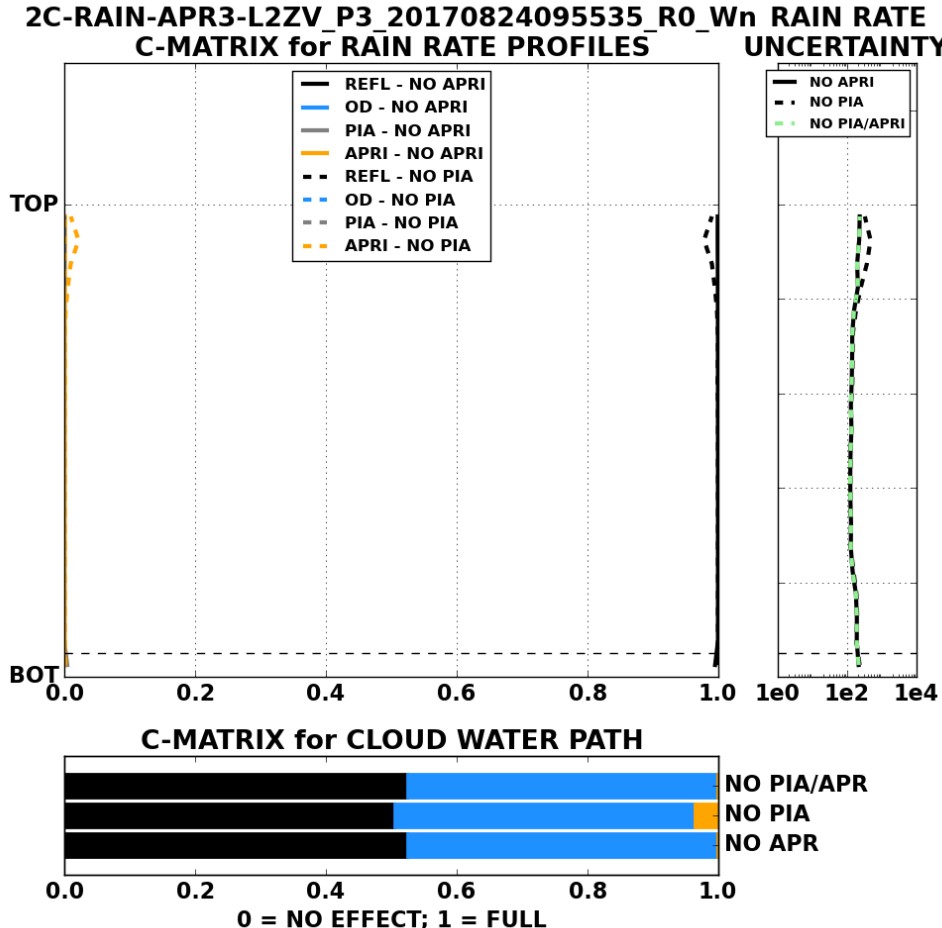

**Figure 4:** As in Fig. 3, but for profiles where the a priori constraint contributed minimal uncertainty (solid line) and profiles where the a priori or PIA constraints contributed minimal uncertainty (dashed lines).

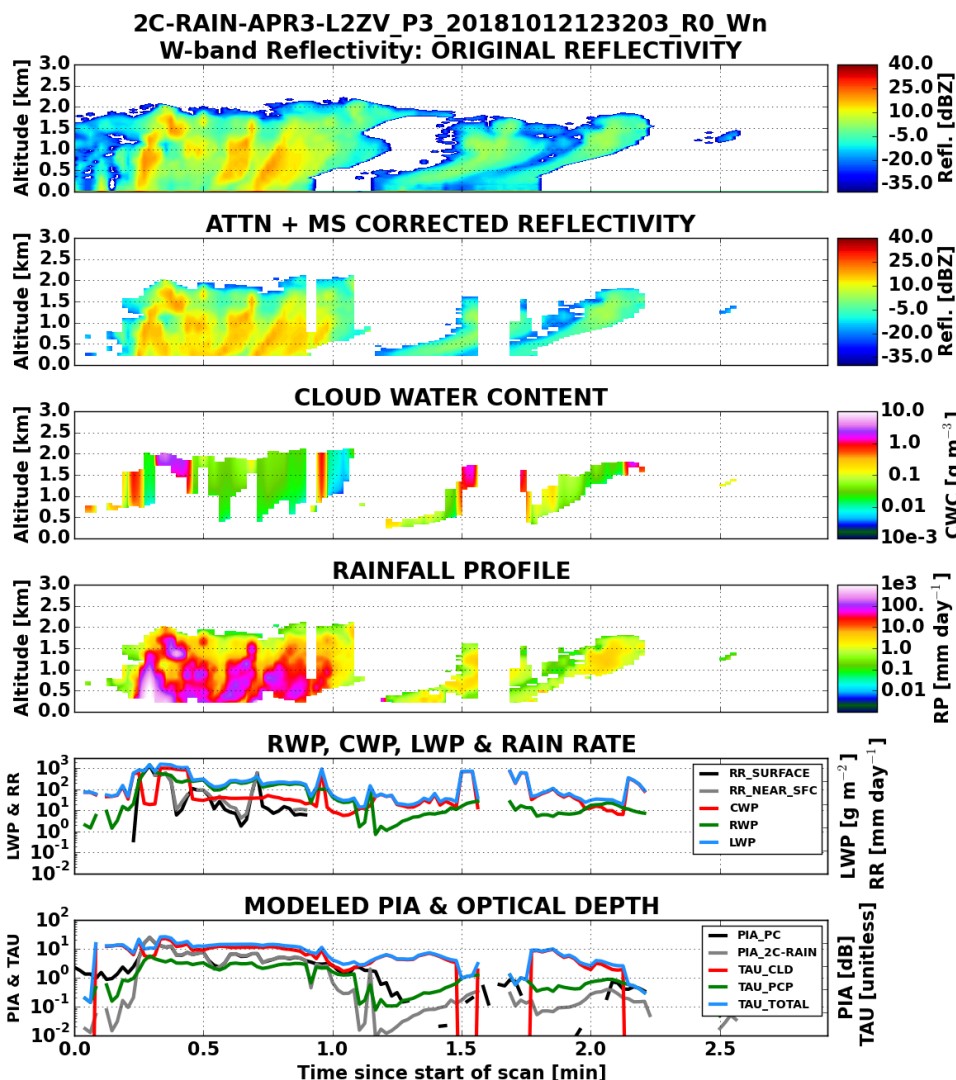

**Figure 5:** Similar to Fig. 2 but for a heavily precipitating trade cumulus cloud observed by the APR-3 on 12 Oct. 2018.





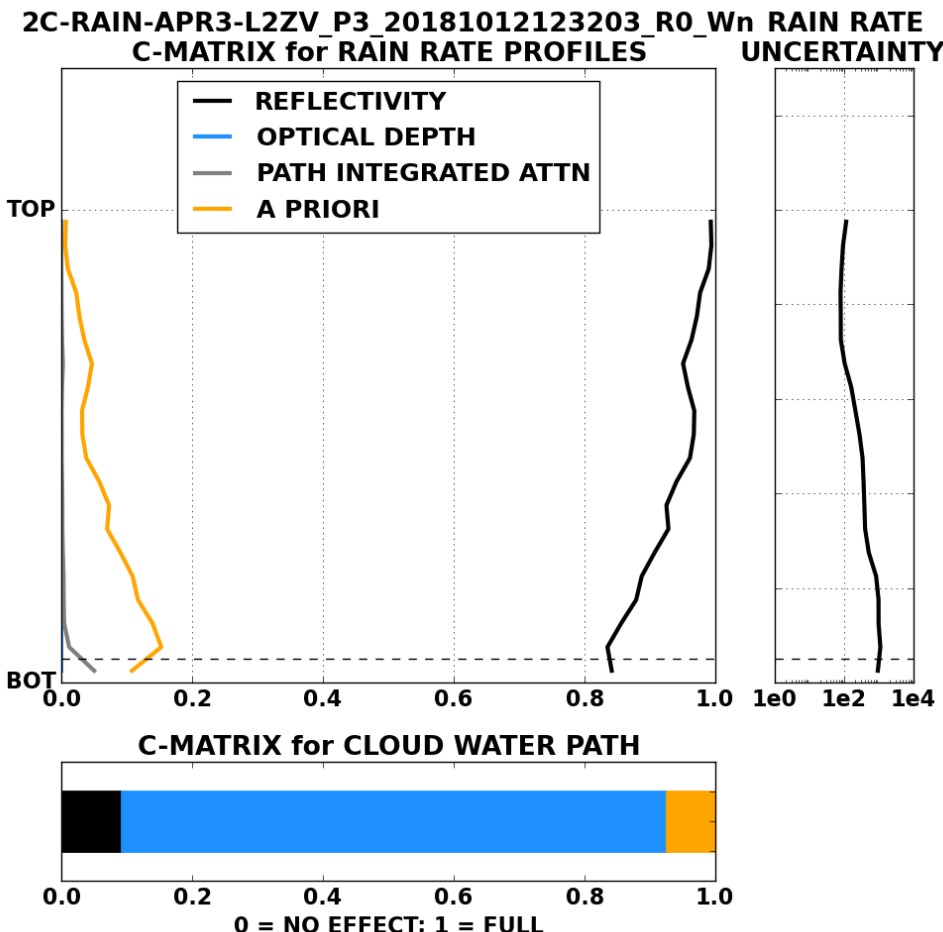

**Figure 6:** Similar to Fig. 3, but for Mean contribution matrix (or C-matrix) calculations relative to cloud

5    height for the radar profiles shown in Fig. 5.





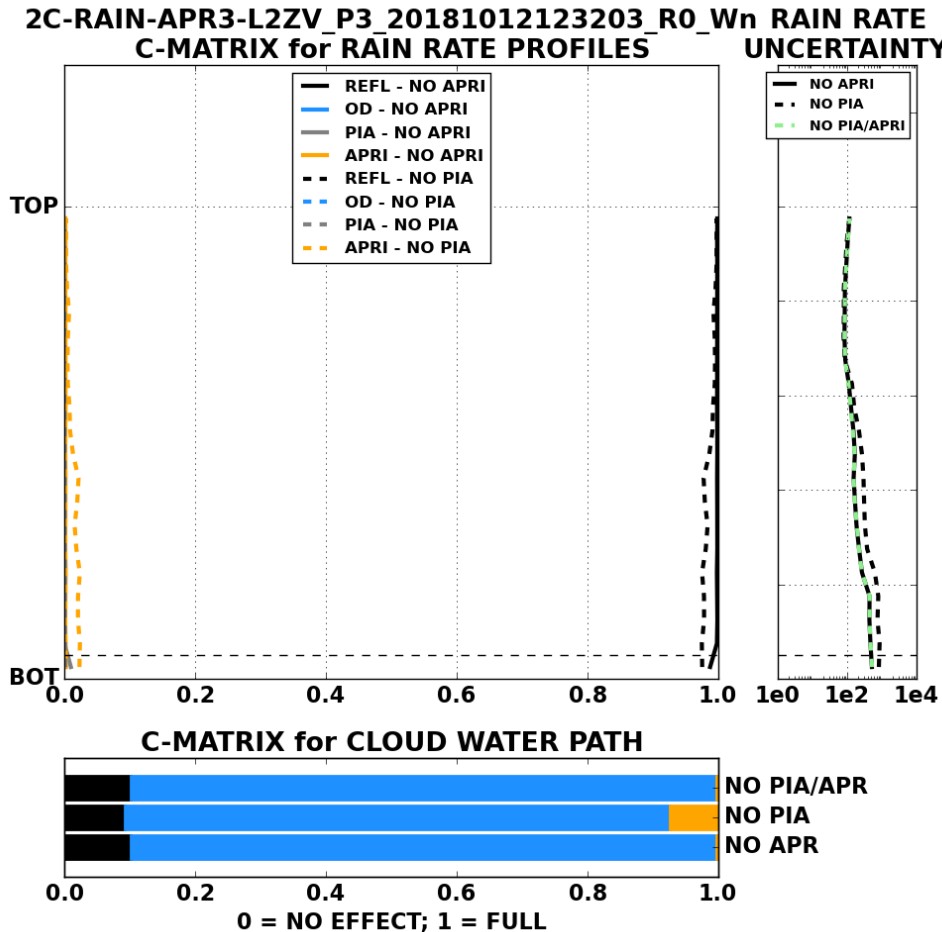

**Figure 7:** As in Fig. 6, but for profiles where the a priori constraint contributed minimal uncertainty (solid line) and profiles where the PIA contributed minimal uncertainty (dashed lines).





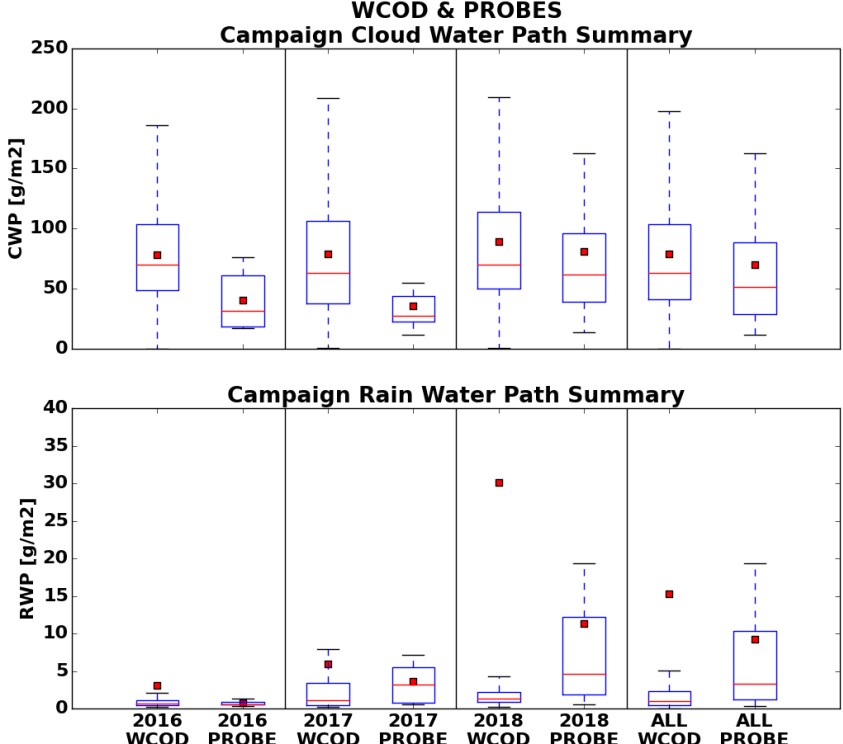

**Figure 8:** A comparison of WCOD derived CWP versus cloud probe estimated CWP (top panel) and RWP (bottom panel). The box and whisker plots represent 95% of all data, and the red squares represent the mean values. In the top panel, WCOD CWP is the combination of all WCOD based retrieved CWPs

5    estimated by the algorithm. Any flight with a missing APR-3, RSP or cloud probe dataset was excluded from the statistics presented for each campaign year.



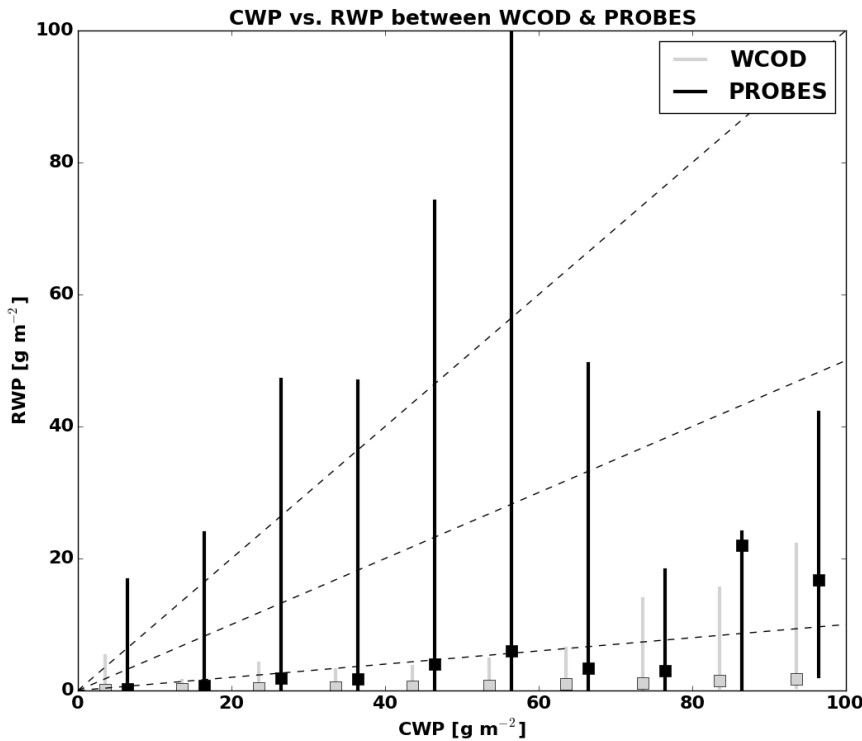

**Figure 9:** Distributions of RWP at various CWP amount, binned every 10 g m$^{-2}$, with boxes representing median values. Each set of lines represents 95% of all data for WCOD (light gray) or cloud probe (black).





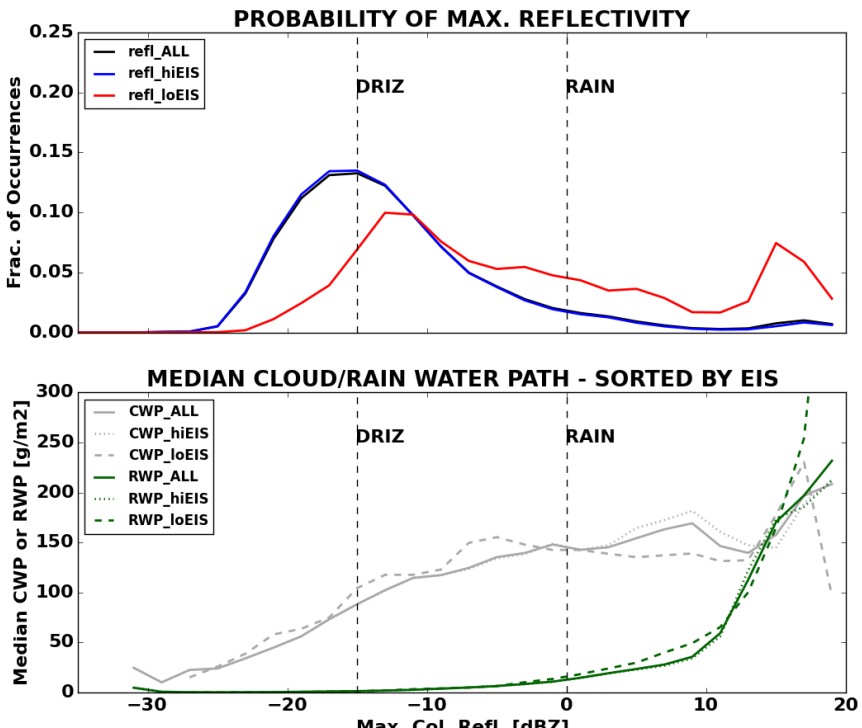

**Figure 10:** The fractional occurrence of maximum column reflectivity in APR-3 detected clouds (top panel) as well as median CWP and RWP values corresponding to maximum column reflectivity bins (bottom panel). Each panel is partitioned further by Estimated Inversion Strength (EIS) denoted by blue (EIS > 0K) and red (EIS < 0K) in the top panel, while CWP and RWP respectively are denoted by dotted (EIS > 0K) and dashed (EIS < 0K) lines respectively.



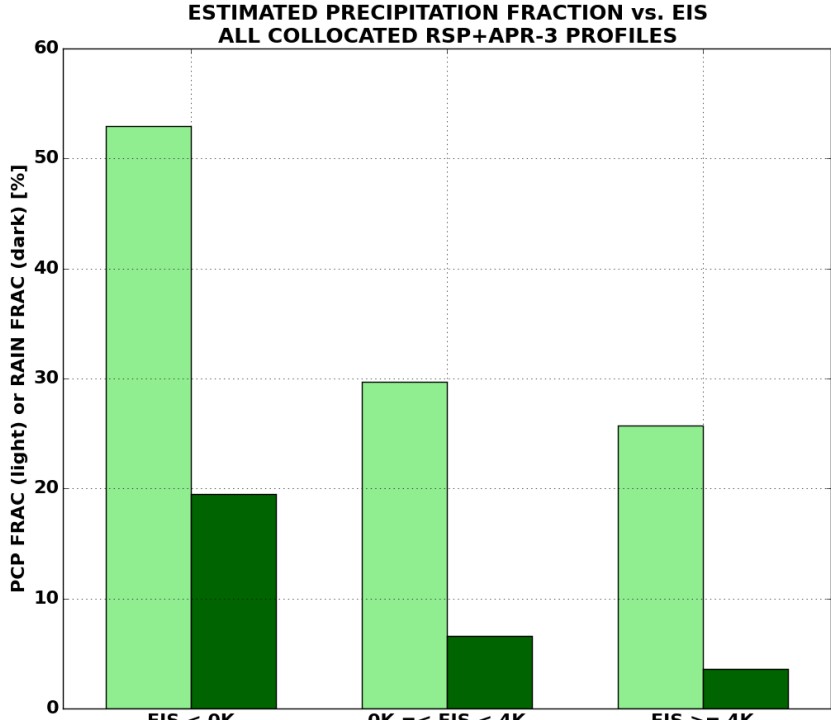

**Figure 11:** Precipitation fraction (light green) and surface rain fraction (dark green) under unstable (EIS < 0K), marginally stable (0 K <= EIS < 4 K) and stable (EIS > 4 K) conditions. Precipitation fraction was estimated from all collocated APR-3 and RSP profiles with a maximum reflectivity of -15 dBZ or larger.

5    Surface precipitation fraction was estimated from any profile where the retrieved surface precipitation rate exceeded 0.01 mm/day.