# Peer review of "Joint Cloud Water Path and Rain Water Path Retrievals from"

_Atmospheric Chemistry and Physics, 2020_

## Referee Comment (RC1) · Anonymous Referee #1 · 2 Sep 2020

Dzambo et al. 2020 Oracles CWP/RWP, Review

The authors have developed a routine that uses multiple aircraft-borne instruments to discern cloud properties in the SE Atlantic Stratocumulus deck, with a focus on partitioning cloud liquid water and rain water. The routine is mostly sensitive for thicker, drizzling stratocumulus, but has a greater amount of uncertainty for more heavily precipitating convective-type clouds. Case studies show reasonable performance, with drizzling stratocumulus clouds containing far more cloud water than rain water, but heavier raining clouds containing more rain water (though results are less uncertain). The routine is stated to be insensitive to aerosols in the environment, making retrievals from this platform ideal for studying the aerosol-cloud interactions in the SE Atlantic.

The manuscript is well explained, cleanly presented, and results are well supported by

the material and figures presented. The RWP/CWP data from ORACLES will provide significant scientific value for the community. The paper is in no need of significant revisions and the routine presented is not particularly controversial as it relies on a great deal of established science. Given the purposes of ORACLES, to specifically study aerosol/cloud interactions, I would like to see a more detailed assurance that the routine is insensitive environmental aerosols. I recommend that the article be accepted in its present form, but with a few small details addressed.

Comment:

Parts of this routine (not the radar part) rely on optical properties of cloud tops. I'm somewhat surprised that overlying aerosols have no effect on cloud bow properties and reflectances. It would benefit the paper to state directly how we can be certain that the routine is insensitive to overlying scattering aerosols, and whether this has been tested. It's possible that this is explained in the referenced material, but a quick explanation here would be beneficial.

Minor Fixes:

For stratocumulus, use the 'Sc' abbreviation.

Page 6: were there any limitations or changes in results when CAS wasn't available and CDP was the only option for n(D)?

At some point around page 7, mu is replaced by u

Line 20/21, page 10, not sure what you mean concerning RWP and CWP, are you implying that the a-priori rate keeps the algorithm from returning unrealistic RWP and CWP?

CWP is basically invisible in Figure 2. This could be by design to highlight how little RWP exists in the cloud compared to CWP, but maybe you could find a way to plot the lines in a semi-transparent way to highlight the overlap?
* * *
Interactive
comment

Line 21 The 2016 ORACLES campaign... Were there any other reasons that the RWP could be lower? Differences in EIS or other environmental variables? This isn't really important for the results, but may be worth looking into.

Figure 8: The text makes it clear, but can you mention in the caption that the obs are quasi-simultaneous?

---

## Referee Comment (RC2) · Anonymous Referee #2 · 16 Oct 2020

The authors have developed an algorithm to joint retrieve cloud water path and rain water path, using W-band reflectivity measurements from aircraft-borne radar, radiometric cloud optical depth, and cloud effect radius from Research Scanning Polarimeter during ORACLES.

Atmospheric Chemistry and Physics (ACP) is a scientific journal dedicated to public discussion of studies investigating the Earth's atmosphere and the underlying chemical and physical processes. In the manuscript, there is no process level discussion. This retrieval algorithm manuscript definitely fits better into the scope of "Atmospheric Measurement Techniques". It is noticed that several retrieval-based papers published in ACP, so a decision would be made by the editor.

The manuscript is well written overall, but more clarifications are greatly needed es-

pecially in the retrieval methodology. How to estimate the cloud effective radius profile is critical in your retrieval. Your assumption for the estimation is not clear. Are you suggesting the cloud top effective radius from RSP is representative of the whole cloud column? Due to cloud-top entrainment, an effective radius at the cloud top can be substantially different from re in the cloud (say, the middle of the cloud layer), see aircraft measurements in Wood (2005). As you will show results for individual cases, it is important to quantify cloud-top entrainment strength and the resulting errors in your retrieval case by case. Otherwise, I see limited values in the retrieval product for detailed case analysis or process analysis. Although the retrievals may be useful for statistic-based study from ORACLES, please prove the values of your retrieval product in process aerosol-cloud interaction study. If these issues are addressed well, the paper might be suitable for publication.

Comments:

I did not see the feasibility of the application of the current retrieval algorithm in the manuscript. If there is no direct observations from aircraft or RSP, how would you apply this algorithm? Further clarification is needed.

Page 7 lines 1-4: When the V channel data was available, are droplet properties similar from 2DS H and V channels? What are the differences in rain droplet size, concentration, and water content from the two channels?

Page 7 lines 8-12: how do you determine if drizzle presents? 2DS can record valid values even in drizzle free regions.

Page 8 line 12: Please briefly describe the parameterization.

Page 10 Eq 10. Should you include the effect of uncertainties of cloud effective radius?

Will Ka-band see part of the cloud with some overlapping regions from W-band observation? Would the different Ka-W band measurements add more information in your retrieval?

---

## Author Comment (AC1) · 14 Dec 2020

**Author Responses to Anonymous Referee #1**

**Anonymous Referee #1**

Dzambo et al. 2020 Oracles CWP/RWP, Review

The authors have developed a routine that uses multiple aircraft-borne instruments to discern cloud properties in the SE Atlantic Stratocumulus deck, with a focus on partitioning cloud liquid water and rain water. The routine is mostly sensitive for thicker, drizzling stratocumulus, but has a greater amount of uncertainty for more heavily precipitating convective-type clouds. Case studies show reasonable performance, with drizzling stratocumulus clouds containing far more cloud water than rain water, but heavier raining clouds containing more rain water (though results are less uncertain). The routine is stated to be insensitive to aerosols in the environment, making retrievals from this platform ideal for studying the aerosol-cloud interactions in the SE Atlantic.

> We thank you very much for taking the time to review and comment on this manuscript! We have responded to each of your comments in blue text (indented after and below each comment), and hope that each of our comments sufficiently addresses each of your concerns. Updates or changes to the manuscript text are denoted in red text.

The manuscript is well explained, cleanly presented, and results are well supported by the material and figures presented. The RWP/CWP data from ORACLES will provide significant scientific value for the community. The paper is in no need of significant revisions and the routine presented is not particularly controversial as it relies on a great deal of established science. Given the purposes of ORACLES, to specifically study aerosol/cloud interactions, I would like to see a more detailed assurance that the routine is insensitive environmental aerosols. I recommend that the article be accepted in its present form, but with a few small details addressed.

> This is an excellent point, and one that we definitely feel needs to be addressed in this manuscript. Ultimately, we do need to demonstrate (or at minimum, clearly state any related limitation) how sensitive our algorithm is to overlying aerosols. In short, the RSP cloud effective radius products are insensitive to overlying aerosol, but the cloud optical depth products are likely somewhat sensitive to overlying aerosol. We have written a more thorough response below, which we believe addresses this concern.

Comment:

Parts of this routine (not the radar part) rely on optical properties of cloud tops. I'm somewhat surprised that overlying aerosols have no effect on cloud bow properties and reflectances. It would benefit the paper to state directly how we can be certain that the routine is insensitive to overlying scattering aerosols, and whether this has been tested. It's possible that this is explained in the referenced material, but a quick explanation here would be beneficial.

Thank you for bringing this to our attention. We were unclear with this important aspect: RSP cloud top effective radius is *not* sensitive to overlying aerosol, but the cloud optical depth is sensitive to overlying aerosol. Aside from the references provided in the RSP methodology sections, we are unaware of any studies evaluating the effect of overlying aerosols on RSP COD retrievals (visible radiances). We have updated the text to ensure these points are very clear.

"Cloud optical depth (COD) is retrieved using **radiometric reflection in the non-absorbing 864 nm band**, which may be affected by overlying aerosols, **and the droplet size retrieval** [cf. Nakajima and King, 1990]. Unlike the reflectance-based COD retrieval, multi-angle polarimetric cloud-top retrievals rely **only** on the shape of the cloudbow, not its intensity, and therefore **is nearly** unaffected by above-cloud aerosol layers and cloud 3-D effects [Alexandrov et al., 2012]."

Minor Fixes:

For stratocumulus, use the 'Sc' abbreviation.

We have taken care to update all former "StCu" abbreviations into "Sc" abbreviations throughout the text.

Page 6: were there any limitations or changes in results when CAS wasn't available and CDP was the only option for n(D)?

This is an interesting question. We did not explicitly check to see how different CAS and CDP n(D) in cases where they were both operating. With this in mind, the CAS and CDP instruments were mounted next to each other on the P-3, and we expect there to be minimal differences. In any case, this question will be investigated further and will ensure by final publication that CAS and CDP n(D) were not significantly different when operating together.

At some point around page 7, mu is replaced by u

We reviewed the text around page 7, and found no mention or use of "mu" or "u" on our part. We thank you for bringing this up regardless.

Line 20/21, page 10, not sure what you mean concerning RWP and CWP, are you implying that the a-priori rate keeps the algorithm from returning unrealistic RWP and CWP?

Thank you for bringing this to our attention. Yes, the entire intent and design of the a priori constraint is to ensure the final retrieved precipitation rates, RWP and CWP are within reasonable boundaries and prevents unrealistic CWP and RWP values from being returned. The last part of this sentence: "while also providing reasonable estimates for each" is inaccurate, and thus we eliminated this part from the text.

CWP is basically invisible in Figure 2. This could be by design to highlight how little RWP exists in the cloud compared to CWP, but maybe you could find a way to plot the lines in a semi-transparent way to highlight the overlap?

> This is a very good suggestion. Indeed, the design of this figure was to optimize visibility as best as possible given there's admittedly a lot of data, and we intended to show that CWP dominates this scene. We will provide an update version of this figure that will show CWP/RWP/total (and rain/cloud optical depths in the bottom panel) much more clearly, either by using different sized lines or semi-transparent lines.

Line 21 The 2016 ORACLES campaign... Were there any other reasons that the RWP could be lower? Differences in EIS or other environmental variables? This isn't really important for the results, but may be worth looking into.

> We were unsure which page you were referring to Line 21, but have done our best to address this question. There are many reasons (we think) RWP could be potentially lower. First, in 2016, a few flights took place very close to the African coast (where boundary layer heights were very low, often less than 800 meters) and a substantial fraction of measurements were taken just off the coast during routine flights. Clouds here were very thin, and hence RWP could reflect sampling of these environments conducive for thinner cloud. Second, EIS indeed is generally a lot higher in these low PBL altitude regimes. Finally, the peak of biomass burning season occurred during the 2016 experiment, so aerosols could very well have played a role (or contributed majorly) explaining why RWP was much lower. These questions are all being investigated, and appreciate your thought-provoking question here!

Figure 8: The text makes it clear, but can you mention in the caption that the obs are quasi-simultaneous?

> We thank you for bringing this to our attention. In order to emphasis the methodology within the text, we followed your suggestion and updated the caption to the following (new addition is italicized here for clarity):

> "A comparison of WCOD derived CWP versus cloud probe estimated CWP (top panel) and RWP (bottom panel). *Quasi-simultaneous WCOD profiles occurring within 5 minutes and 0.125 degrees latitude (~13 km) of an in-situ profile are included in this comparison.* The box and whisker plots represent 95% of all data, and the red squares represent the mean values. In the top panel, WCOD CWP is the combination of all WCOD based retrieved CWPs estimated by the algorithm. Any flight with a missing APR-3, RSP or cloud probe dataset was excluded from the statistics presented for each campaign year."

> Upon reviewing this portion of the text, the code generating Fig. 8 actually used a lat/lon threshold of 0.125 degrees, and thus updated the text to reflect this correction (not 0.25 degrees as originally written and implied).

---

## Author Comment (AC2) · 14 Dec 2020

**Author Responses to Anonymous Referee #2**

**Anonymous Referee #2**

The authors have developed an algorithm to joint retrieve cloud water path and rain water path, using W-band reflectivity measurements from aircraft-borne radar, radiometric cloud optical depth, and cloud effect radius from Research Scanning Polarimeter during ORACLES.

We thank you very much for taking the time to review our work and provide constructive comments toward improving this manuscript! We have responded to each of your comments in blue text (indented after and below each comment), and hope that each of our comments sufficiently addresses each of your concerns. Updates or changes to the manuscript text are denoted in red text.

Atmospheric Chemistry and Physics (ACP) is a scientific journal dedicated to public discussion of studies investigating the Earth's atmosphere and the underlying chemical and physical processes. In the manuscript, there is no process level discussion. This retrieval algorithm manuscript definitely fits better into the scope of "Atmospheric Measurement Techniques". It is noticed that several retrieval-based papers published in ACP, so a decision would be made by the editor.

We understand the point you are trying to make here. We feel, however, that the content of this manuscript goes beyond the typical scope of an AMT paper – especially the results in Section 6. The manuscript blends algorithm details and analysis that supports cloud & precipitation processes in the aerosol-rich southeast Atlantic environment. Ultimately, this manuscript will be part of an inter-journal ACP/AMT special issue "New Observations and related modeling studies of the aerosol-cloud-climate system in the Southeast Atlantic and Southern Africa regions" and feel that ACP is the more appropriate choice given the scope. Several process-level studies using the dataset presented in this work are planned, and will be the topic of future manuscripts within this special issue.

In short, we feel the submission is justified for ACP but ultimately would respect the editor's decision to either keep the manuscript here in ACP or transfer the manuscript to AMT.

The manuscript is well written overall, but more clarifications are greatly needed especially in the retrieval methodology. How to estimate the cloud effective radius profile is critical in your retrieval. Your assumption for the estimation is not clear. Are you suggesting the cloud top effective radius from RSP is representative of the whole cloud column? Due to cloud-top entrainment, an effective radius at the cloud top can be substantially different from re in the cloud (say, the middle of the cloud layer), see air- craft measurements in Wood (2005). As you will show results for individual cases, it is important to quantify cloud-top entrainment strength and the resulting errors in your retrieval case by case. Otherwise, I see limited values in the retrieval product for detailed case analysis or process analysis. Although the retrievals may be useful for statistic-based study from ORACLES, please prove the values of your retrieval product in process aerosol-cloud interaction study. If these issues are addressed well, the paper might be suitable for publication.

The cloud effective radius assumption, we argue, is not a critical component to our retrieval. This is because the cloud water path (CWP) is a bulk quantity and relies mostly on the cloud top effective radius. We certainly agree and acknowledge that processes such as entrainment, accretion and autocorrelation will vary CWC on a profile-by-profile basis depending on the point in the cloud profile's lifetime. Cloud water content (CWC) through each individual profile is assumed following the Bennartz (2007) method, whereby integrating CWC in each profile will equal the CWP regardless of how effective radius varies through the column. We did not make clear, however, that uncertainty arising from effective radius through the profile is not accounted for, and added the following sentence within Line 6 on Page 10:

"Given this assumption for CWC through the profile, we do account for variability in $r_e$ through the profile and uncertainty that may arise from variable $r_e$ through each profile."

With this in mind, future studies will be addressing in-situ cloud properties (cloud water content, effective radius, etc.) much more rigorously and will provide a basis for further evaluation and improvement of the treatment of CWC in this algorithm. Furthermore, studies covering the effects of entrainment will be explored in future studies. The entire premise of this algorithm is to provide bulk precipitation, CWP and RWP estimates (each constrained by radar and radiometric measurements) for future statistic-based and process-level aerosol-cloud-precipitation interaction studies.

The science behind this specific algorithm is very well established through numerous previous studies, and is the only algorithm (and corresponding data product) available for ORACLES that encapsulates both cloud and precipitation processes using all valid and available W-band radar and polarimetric radiometer data.

Comments:

I did not see the feasibility of the application of the current retrieval algorithm in the manuscript. If there is no direct observations from aircraft or RSP, how would you apply this algorithm? Further clarification is needed.

This algorithm was developed from satellite based (CloudSat + MODIS) algorithm methodologies for retrieving rainfall profiles and cloud properties. This algorithm was certainly optimized for airborne remote sensing observations (APR-3 + RSP), and can be applied to any airborne platform field campaign using radar & radiometers. We perhaps should have clarified in the title that this algorithm is meant for airborne application. We will update the title of the manuscript to specify "Airborne", as in:

**Joint Cloud Water Path and Rain Water Path Retrievals from Airborne ORACLES Observations**

Page 7 lines 1-4: When the V channel data was available, are droplet properties similar from 2DS H and V channels? What are the differences in rain droplet size, concentration, and water content from the two channels?

The figures below compare the droplet concentration sampled by the horizontal (NH) and vertical (NV) channels of the 2-Dimensional Stereo probe (2-DS). Each data point on the figures represents a 1 Hz measurement colored by the effective radius (Re). For the 2017 deployment, NH and NV were compared for 6125 data points from eight research flights. For the 2018 deployment, NH and NV were compared for 9886 data points from twelve research flights. Statistically significant (p = 0) linear fit coefficients were derived using a linear regression model, and NV = 0.91 NH + 0.0033 for 2017, and NV = 0.96 NH - 0.0017 for 2018. NH and NV were well correlated with Pearson's correlation coefficient, R = 0.94 for 2017 and R = 0.98 for 2018.

A more comprehensive analysis of 2-DS precipitation measurements and properties will be the topic of a future paper, and thus we elected not to include the following figures in the text. Instead we elaborated at this point in the text to justify our use of H-channel data only:

"This decision is justified by the fact that available n(D) data between the H and V channels were highly correlated. The Pearson correlation coefficients between the droplet concentration (Nc) were 0.94 (N = 6125) and 0.98 (N = 9886) for 2017 and 2018 respectively."

[Figure]

[Figure]

Page 7 lines 8-12: how do you determine if drizzle presents? 2DS can record valid values even in drizzle free regions.

      Droplets larger than 50 μm in diameter were defined as drizzle. The presence of drizzle was determined when 2-DS registered droplets within the size bins corresponding to this size range.

Page 8 line 12: Please briefly describe the parameterization.

      This parameterization (Equation 4) is a model for visible optical depth. We did not explicitly state this prior to presenting this equation (but do so later around Line 18). To make this more clear, we added additional context on Line 11:

      "… The visible optical depth observed by the RSP includes contributions from CWC and RWC, *and can be modeled as (also see Lebsock and L'Ecuyer, 2011):*"

Page 10 Eq 10. Should you include the effect of uncertainties of cloud effective radius?

      We only included the uncertainty in cloud effective radius following our assumption. There will certainly be variation in cloud effective radius through the column, however, given the nature of our retrieval algorithm there is no need to account for effective radius through the column. The column cloud water content assumption, which also requires the assumed cloud effective radius through the column, will likely be covered in future ORACLES papers and will provide further evaluation of the validity of this assumption.

Will Ka-band see part of the cloud with some overlapping regions from W-band observation? Would the different Ka-W band measurements add more information in your retrieval?

The Ka-band channel is insensitive to cloud size droplets (i.e. less than –15 dBZ), but will usually detect precipitation when it's present. The development of a Ka-W band joint channel retrieval was actually the original idea for retrieval development, however, the W-band + cloud optical depth retrieval was developed because nearly all stratocumulus clouds (with exceptions for trade cumulus and similar cases like the one presented in Fig. 5) were fully detectable by the W-band channel. Ka-band measurements would add much more information in cases such as those similar to Fig. 5, since the Ka-band can penetrate deeper into more heavily precipitating clouds before (if at all) partially or fully attenuating.